# Attention-related modulation of caudate neurons depends on superior colliculus activity

**James P Herman[1]\*, Fabrice Arcizet[2], Richard J Krauzlis[1]\***

[1]Laboratory of Sensorimotor Research, National Eye Institute, Bethesda, United States; [2]INSERM, CNRS Institut de la Vision, Paris, France

**Abstract** Recent work has implicated the primate basal ganglia in visual perception and attention, in addition to their traditional role in motor control. The basal ganglia, especially the caudate nucleus 'head' (CDh) of the striatum, receive indirect anatomical connections from the superior colliculus (SC), a midbrain structure that is known to play a crucial role in the control of visual attention. To test the possible functional relationship between these subcortical structures, we recorded CDh neuronal activity of macaque monkeys before and during unilateral SC inactivation in a spatial attention task. SC inactivation significantly altered the attention-related modulation of CDh neurons and strongly impaired the classification of task-epochs based on CDh activity. Only inactivation of SC on the same side of the brain as recorded CDh neurons, not the opposite side, had these effects. These results demonstrate a novel interaction between SC activity and attention-related visual processing in the basal ganglia.

## Introduction

Covert visual attention is the ability of primates to selectively make use of some visual inputs while ignoring the rest, without moving the eyes. It confers exquisite behavioral and cognitive flexibility. For example, a male macaque of lower social rank can simultaneously try to appease a higher-ranking male by avoiding direct eye contact and monitor the higher-ranking male's behavior for signs of overt aggression. In laboratory tests, covert attention is reliably associated with changes in the speed and accuracy of behavioral reports (*Carrasco, 2011*). The neural mechanisms for covert visual attention in primates include the modulation of neuronal activity in visual cortical areas that represent stimulus features (*Reynolds and Chelazzi, 2004*; *Treue, 2001*), as well as areas of the frontal and parietal cortex that regulate what is attended (*Bisley and Goldberg, 2010*; *Moore and Zirnsak, 2017*). More recently, it has been recognized that in addition to these cortical mechanisms, the control of covert attention also includes subcortical brain regions, consistent with the idea that covert attention in primates evolved from older subcortical circuits responsible for overt orienting movements triggered by stimulus events (*Krauzlis et al., 2018*).

One of the most important subcortical structures for the control of covert visual attention in primates is the superior colliculus (SC), located in the midbrain. When SC neuronal activity is perturbed by inactivation or microstimulation, performance in attention tasks is reliably altered in a spatially specific manner, even during covert tasks (*Bogadhi et al., 2019*; *Bollimunta et al., 2018*; *Cavanaugh and Wurtz, 2004*; *Herman et al., 2018*; *Lovejoy and Krauzlis, 2010*; *Müller et al., 2005*; *Zénon and Krauzlis, 2012*). During SC inactivation, improvements in perceptual sensitivity made possible by spatial cues are abolished (*Lovejoy and Krauzlis, 2017*). Despite SC inactivation preventing spatial cues from conferring a perceptual advantage, cue-related modulation of neuronal activity in extrastriate visual cortex remains robust during inactivation (*Zénon and Krauzlis, 2012*). Specifically, *Zénon and Krauzlis, 2012* found that in a covert motion-change detection task, neurons

**\*For correspondence:**
hermanj@gmail.com (JPH);
richard.krauzlis@nih.gov (RJK)

**Competing interests:** The authors declare that no competing interests exist.

in visual areas MT and MST showed the same cue-related modulation during SC inactivation as they had before. Because behavior in this task depends on signals arising from MT/MST, these results suggest that SC inactivation impairs behavior by altering the use of cortical sensory signals in another brain area. One candidate site, based on the convergence of signals from visual cortex and the SC, is the striatum of the basal ganglia (*Redgrave, 2010*; *Saint-Cyr et al., 1990*), in particular, the 'head' of the caudate nucleus (CDh).

The caudate is a primary input nucleus of the basal ganglia with distinct territories that have been implicated in value-based decision making, perceptual choices and selective attention. The caudate is divided into the 'head' (CDh) at the anterior end, followed by the 'body', 'genu', and 'tail' (CDt). In keeping with the parallel functional circuit architecture of the basal ganglia (*Alexander et al., 1986*), these territories appear to have distinct functional roles. For example, inactivating CDh neurons impairs choice among visual stimuli that are flexibly associated with high or low reward but does not impair choice of stimuli with fixed reward association, whereas CDt inactivation impairs only choices with fixed reward association stimuli and not flexible reward (*Kim and Hikosaka, 2013*). Microstimulation of CDh and anterior caudate body neurons spatially biases perceptual decisions and alters decision times with random-dot-motion stimuli, consistent with a role for CDh neurons in linking cortical visual signals, perceptual choice, and spatial selection (*Ding and Gold, 2012*). Finally, during an attention task requiring monkeys to perform covert perceptual judgments, CDh and body neurons are strongly modulated by the location of a spatial cue, response choice, or both (*Arcizet and Krauzlis, 2018*). These results suggest that CDh neuronal activity is driven by a combination of sensory signals, information about behavioral relevance, spatial location, and response choice.

We hypothesized that activity in the intermediate and deep layers of primate SC contributes directly or indirectly to cue-related modulation of neuronal activity in the caudate nucleus. To test this idea, we recorded the activity of populations of CDh neurons with a pair of linear electrode arrays while monkeys performed a covert attention task – both before and during unilateral inactivation of the SC. Our results demonstrate that inactivation of SC on the 'same side' of the brain as recorded CDh neurons changes how those neurons represent stimulus relevance while monkeys are performing covert perceptual judgments. We find that same-side SC inactivation: (1) causes clear shifts in the cue-side preferences of CDh neurons; (2) disrupts the ability of a classifier to uniquely identify distinct task-epochs on the basis of CDh neuronal activity; and (3) alters the structure of correlations in CDh neuron populations, consistent with reducing the influence of a common input signal. Our results demonstrate a causal link from the SC to the basal ganglia that could alter how sensory signals are used to guide perceptual choices without altering the sensory representations in visual cortex.

## Results

To measure the effects of superior colliculus (SC) inactivation on neuronal activity in the CDh, we recorded the activity of CDh neurons while monkeys performed an attention task both before and during unilateral SC inactivation. In each experimental session, data were first collected over several blocks of trials before SC inactivation, muscimol was then injected into SC and the presence of a contralateral saccade deficit was confirmed, and finally data were collected over several additional blocks during the effects of inactivation (*Figure 1A*). Because the SC output with access to caudate is almost totally ipsilateral (*Grofov, 1979*; *Harting et al., 1980*; *Matsumoto et al., 2001*; *May et al., 2009*; *Nakano et al., 1990*; *Partlow et al., 1977*), in a majority of sessions we inactivated SC on the same side of the brain as recorded CDh neurons (n = 9 'same-side' inactivations). We also collected data in several 'opposite-side' SC inactivation plus recording sessions (n = 4).

Neuronal and behavioral data were collected while monkeys performed a covert motion-change detection task. Two monkeys (P and R) were trained to release a joystick in response to a direction-of-motion change at a cued location and withhold their response if the change happened at a foil location (*Figure 1B*). Monkeys obtained liquid reward either by responding to a cued motion change with a joystick release (a hit), or by withholding response to a foil change (a correct reject); no reward was given if the monkey failed to respond to a cued change (a miss) or responded to a foil change (a false alarm).

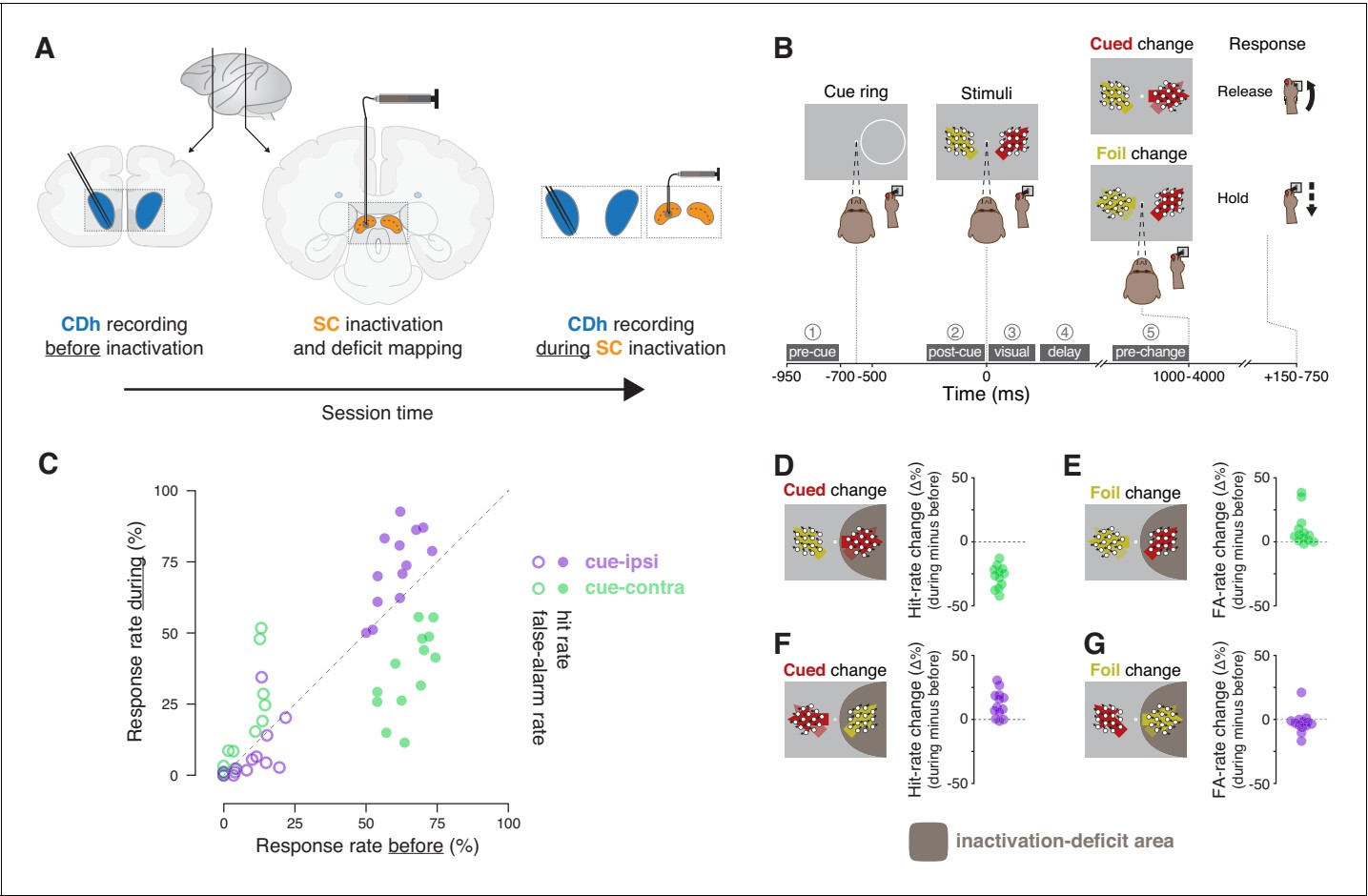

**Figure 1.** Experimental approach and behavioral effects of inactivation. (A) In each session, the activity of neurons in the head of the caudate nucleus (CDh; blue shaded regions) was recorded with a pair of linear 32-channel probes before inactivation. An injection cannula was advanced into SC, 0.5 μL of muscimol was injected, and the presence of a contralateral saccade deficit was confirmed (the images depict an inactivation of SC on the same side of the brain as recorded CDh neurons; not depicted are opposite-side SC inactivations). A diagram of a lateral view of a macaque brain shows the antero-posterior positions (vertical black lines with adjoining arrows) corresponding to the diagrams of coronal slices displaying CDh and SC. During the effects of SC inactivation (referred to throughout as 'during inactivation' or simply 'during'), recordings of the activity of CDh neurons continued. (B) Monkeys performed a cued motion-direction change detection task while CDh neuron activity was recorded before and during SC inactivation. Liquid reward was obtained either by responding to a cued motion change with a joystick release (a hit), or by withholding response to a foil change (a correct reject); no reward was given if the monkey failed to respond to a cued change (a miss) or responded for a foil change (a false alarm). Dark-gray boxes with white text with circled numerals above show the timing, names, and numerical indexes of 'task-epochs' used for data analysis. (C) Performance summary comparing before and during SC inactivation. Hit (filled symbols) and false-alarm rates (empty symbols) during SC inactivation are plotted against rates before SC inactivation from the same sessions (n = 13). Rates for cue-ipsilateral to SC inactivation site are plotted in purple and those for cue-contralateral are plotted in green. (D–G) Differences in hit and false-alarm rates (during minus before SC inactivation) are plotted for each session, horizontal symbol spacing is artificially jittered to increase visibility. Brown shaded region depicts the area of visual space affected by muscimol inactivation of SC.

The online version of this article includes the following figure supplement(s) for figure 1:

**Figure supplement 1.** Performance summary comparing before and during SC inactivation broken down by SC inactivation side and subject.

**Figure supplement 2.** Inactivation-deficit area maps for all recording plus inactivation sessions (n = 13).

**Figure supplement 3.** Electrode contact locations.

## Effects of SC inactivation on attention-task performance

SC inactivation reliably produced spatially specific impairments in attention-task performance (*Figure 1C*; *Figure 1—figure supplement 1*). During each SC inactivation, one of the two stimulus locations was entirely within the area of the visual field affected by muscimol injection (*Figure 1—figure supplement 2*). Consistent with previous reports (e.g. *Zénon and Krauzlis, 2012*), when the

cue was presented inside the inactivation-deficit area of the visual field (cue-contra), hit rate decreased (*Figure 1D*) and false-alarm rate increased (*Figure 1E*) relative to performance before inactivation. When the cue-ring was outside the deficit area (cue-ipsi), hit rate increased (*Figure 1F*) and false-alarm rate showed little change (*Figure 1G*). We quantified the effects of inactivation on performance by comparing hit and false-alarm rates before to during SC inactivation with $\chi^2$-proportion tests (*Fleiss et al., 2013*), which confirmed that cue-contra hit rate during SC inactivation was significantly reduced in each session (13/13 sessions; all $\chi^2$ >9.5, all p 0.01). These tests also indicated that cue-ipsi hit rate increased significantly in 5/13 sessions (all $\chi^2$ >7.1, all p < 0.01).

## Effects of SC inactivation on CDh cue location preferences

To examine the effects of SC inactivation on CDh neuronal activity, we used a pair of 32-channel linear probes in each 'inactivation plus recording' session. This allowed us to make efficient use of each inactivation session, yielding 281 neurons identified as putative medium spiny neurons (MSNs, hereafter referred to as 'CDh neurons'; 171 from monkey P and 110 from monkey R), collected across 13 inactivation sessions (nine same-side: five in monkey P and four in monkey R; four opposite-side: two each in monkeys P and R). CDh neuron data from the two monkeys were pooled for subsequent analyses. We determined the effects of SC inactivation on CDh neuronal responses in two ways. First, we used established methods for tracking individual neurons across recording sessions (*Eleryan et al., 2014*; *Fraser and Schwartz, 2012*) to identify a 'continuously isolated' subpopulation of 194/281 CDh neurons that were most likely to be identical across before and during phases. Second, we treated data collected before and during inactivation as independent subpopulations, making no assumptions to track the identity of CDh neurons across before and during phases, and maximizing neuronal yield. As we describe below, both approaches gave the same results.

Many CDh neurons were robustly modulated by the location of the spatial cue, and these preferences changed during SC inactivation in different ways depending on whether we inactivated SC on the same side of the brain as the CDh recording site or the opposite side. Before inactivation, a 'continuously isolated' example neuron showed greater activity for cue presentation in the visual field contralateral to CDh recordings (cue-contra) than the ipsilateral visual field (cue-ipsi), illustrating a cue-contra preference (*Figure 2A*). During same-side SC inactivation, the example neuron instead exhibited a modest cue-ipsi preference which was most apparent immediately preceding the direction-of-motion change (*Figure 2B*; 'pre-change' task-epoch). To quantify cue-side preferences of individual neurons, we computed the area under a receiver operating characteristic curve (ROC area; *Britten et al., 1992*) in each of several temporal epochs defined by task events (1: pre-cue, 2: post-cue, 3: visual, 4: delay, 5: pre-change; *Figure 1B*), as well as across task-epochs. Before same-side SC inactivation, the example unit (in 2A) had significant cue-contra preferences in epochs 1–4 (*Figure 2C*; all permutation-test 95% confidence intervals greater than 0.5). During same-side SC inactivation, the example unit (in 2B) had no significant cue-side preference in epochs 1–4 (all 95% ROC CIs ⊂ 0.5), and a significant cue-ipsi preference in epoch 5 (95% ROC CI <0.5; *Figure 2D*). Thus same-side SC inactivation markedly reduced the example neuron's cue-contra preference.

In contrast, opposite-side SC inactivation softly biased cue-side preferences towards cue-contra. Another 'continuously isolated' example neuron had significant cue-ipsi preferences across epochs (*Figure 2E*; all ROC CIs < 0.5); because we only inactivated one side of SC in each experiment, this neuron was necessarily recorded during a separate session from the example neuron in *Figure 2A–D*. During opposite-side SC inactivation, this example neuron showed weaker cue-ipsi preferences (*Figure 2F*) which remained significant in epochs 2, 3 ,and 5 (*Figure 2H*, ROC CIs < 0.5). Together, these two example neurons illustrate the main pattern in our results – unilateral SC inactivation altered CDh cue-side preferences, with the clearest effect being an attenuation of CDh cue-contra preferences by same-side SC inactivation.

Across our subpopulation of 'continuously isolated' CDh neurons, same-side SC inactivation redistributed cue-side preferences in favor of cue-ipsi and opposite-side SC inactivation weakly pushed preferences towards cue-contra, independent of task-epoch. After determining that neither the prevalence nor the strength of cue-side preferences varied significantly with task-epoch (*Figure 3—figure supplement 1A–F*; prevalence: logistic regression, smallest p = 0.26; strength: ANOVAs, smallest p = 0.64; see methods for details), we summarized the effect of SC inactivation on a single-neuron basis by computing the cue-side preference (ROC area) across task-epochs, before and during inactivation for each neuron (*Figure 3*, *Figure 3—figure supplement 2*; see *Figure 3—figure*

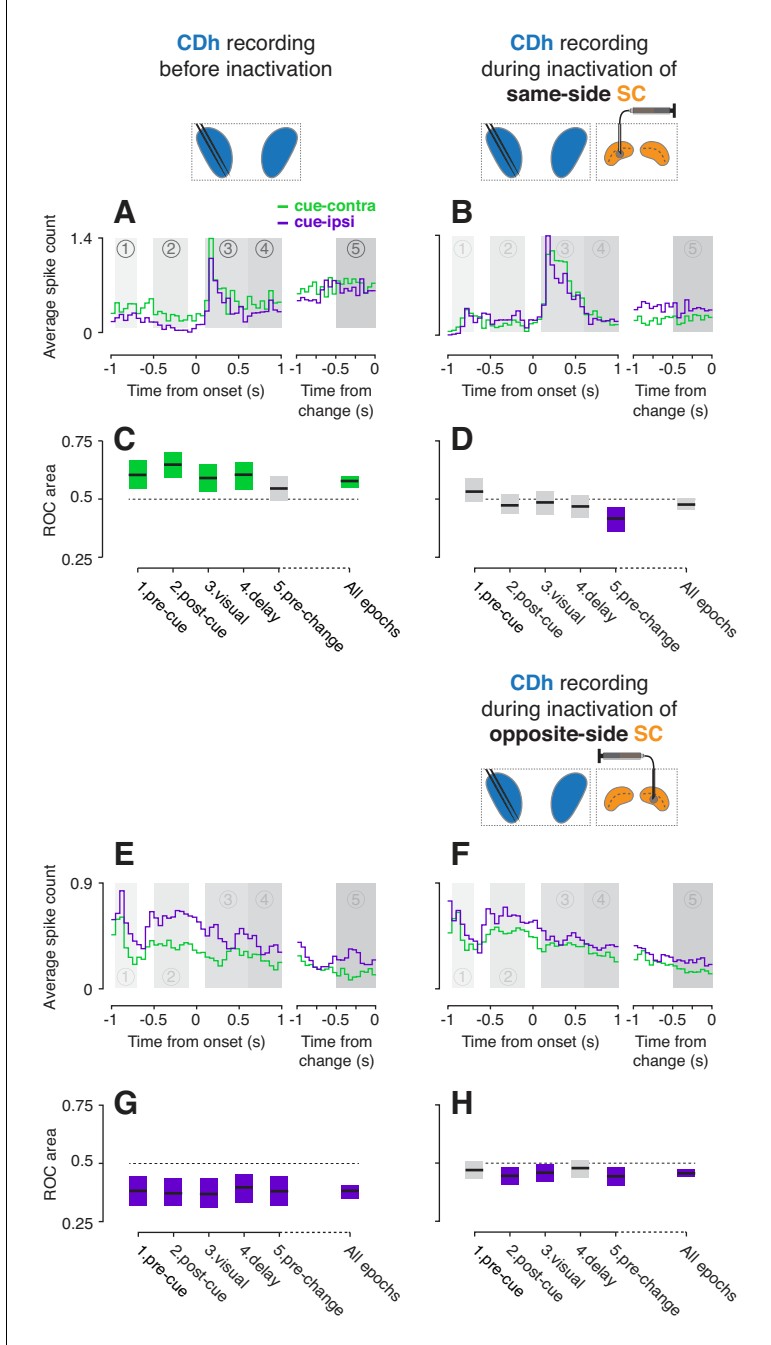

**Figure 2.** Example CDh neurons before and during SC inactivation. (**A**) Activity of an example 'continuously isolated' CDh neuron recorded before same-side SC inactivation. Traces representing average spike counts in non-overlapping 50 ms bins are plotted separately for cue-contralateral (to CDh recordings) and cue-ipsilateral conditions. A portion of data aligned on stimulus onset is presented in the left panel, and a portion aligned on stimulus change in the right panel. Shaded areas mark five task-epochs: 1: pre-cue, 2: post-cue, 3: visual, 4: delay, 5: pre-change. (**B**) Presentation as in panel (**A**) but for activity recorded during same-side SC inactivation. (**C**) The area under a receiver operating characteristic curve (ROC area) was used to quantify the cue-side preference of the example unit in (**A**) within each task-epoch and across all task-epochs ('all epochs' tic label). Horizontal black line segments are ROC areas and surrounding shaded regions indicate bootstrapped 95% confidence intervals (CIs). Individual ROC areas are considered significant when their 95% CIs ⊄ 0.5 (do not contain 0.5; dotted line); significant cue-contra ROC CIs are colored green and significant cue-ipsi CIs are colored purple, gray ROC CIs indicate non-significant cue-side preferences. (**D**) ROC areas and 95% CIs comparing cue-contra to cue-ipsi for the example neuron in panel (**B**). (**E**) Presentation as in panel (**A**) but for an example

*Figure 2 continued on next page*

*Figure 2 continued*

'continuously isolated' unit recorded before opposite-side SC inactivation. (F) Presentation as in panel (E) but for activity recorded during opposite-side SC inactivation. (G) ROC areas and 95% CIs comparing cue-contra to cue-ipsi for the example neuron in panel (E). (H) ROC areas and 95% CIs comparing cue-contra to cue-ipsi for the example neuron in panel (F).

*supplements 3–4* for monkey-specific data). Before same-side SC inactivation 45% of CDh neurons (63/141) had significant cue-contra preferences and 16% (23/141) had significant cue-ipsi preferences (*Figure 3A*; all ROC CIs ⊄ 0.5). During same-side SC inactivation, just 16% (23/141) had significant cue-contra preferences and 34% (48/141) had significant cue-ipsi preferences (*Figure 3B*; all ROC CIs ⊄ 0.5); proportions of both cue-contra and cue-ipsi preferences during same-side SC inactivation were significantly different from those proportions before inactivation ($\chi^2$ proportions tests; cue-contra preferences: $\chi^2$-stat = 27.9424, p < 0.01; cue-ipsi preferences: $\chi^2$-stat = 9.3793, p < 0.01). In contrast, opposite-side SC inactivation increased the prevalence of significant cue-contra preferences from 28% to 36% (15/53 to 19/53; $\chi^2$-stat = 0.6928, p = 0.41) and left cue-ipsi unchanged at 19% (10/53; $\chi^2$-stat = 0, p = 1; *Figure 3—figure supplement 2A,B*; all ROC CIs ⊄ 0.5).

The change in cue-side preferences caused by SC inactivation did not result from a unilateral shift, but rather from a biased redistribution of preferences. To clarify how SC inactivation altered cue-side preferences in continuously isolated CDh neurons, we separated CDh neurons into three groups according to their preferences before inactivation (cue-contra, cue-ipsi, or no-preference; *Figure 3C, E,G*), and visualized each group's distribution of preferences during inactivation (*Figure 3D,F,H*). In each group, regardless of cue-side preferences before inactivation, distributions of cue-side preferences during inactivation contained neurons with cue-contra preferences, cue-ipsi preferences, and no-preference. For example, of the 63 neurons with a significant cue-contra preference before same-side SC inactivation (*Figure 3C*), 22% (14/63) retained cue-contra preference, 22% switched to cue-ipsi, and the remaining 56% (35/63) had no-preference during inactivation. The same kind of biased redistribution of cue-side preferences was also evident, though weaker, in opposite-side SC inactivations (*Figure 3—figure supplement 2*). These results indicate that the loss of SC activity during inactivation reduced the overall prevalence of cue-contra preferences but did not simply shift the preferences of each neuron.

In addition to analyzing this 'continuously isolated' subpopulation on a single-neuron basis, we also examined cue-side preferences by treating the data as 'independent subpopulations' of CDh neurons. Our continuously isolated subpopulation was selected using reasonable established methods but involves assumptions about the identity of individual neurons based on their firing characteristics. Treating neuronal data collected before and during SC inactivation as independent subpopulations eliminates those assumptions.

Analysis of the independent subpopulations of CDh neurons validated our findings from the continuously isolated single neurons. Cue-side preferences in the independent subpopulations of CDh neurons were redistributed towards cue-ipsi by same-side SC inactivation, and weakly towards cue-contra by opposite-side inactivation. Same-side SC inactivation decreased the proportion of CDh neurons with significant cue-contra preferences from 25% to 10% and increased cue-ipsi preferences from 15% to 25% (*Figure 4A,B,E,F*; all ROC CIs ⊄ 0.5; see *Figure 4—figure supplements 1* and *2* for monkey-specific data). Opposite-side SC inactivation increased the proportion of significant cue-contra preferences from 13 to 18% and decreased cue-ipsi from 17 to 13% (*Figure 4C,D,G,H*; all ROC CIs ⊄ 0.5; all percentages collapsed across task-epochs).

To statistically test these changes in proportions of cue-side preferences of independent CDh subpopulations during SC inactivation, we fit proportions in each task-epoch, before and during inactivation, with logistic regression (separately for same-side and opposite-side SC inactivation data). This analysis confirmed that same-side inactivation significantly reduced the proportion of cue-contra preferring CDh units (tStat = −3.2336, p < 0.01) and increased the proportion preferring cue-ipsi (interaction term; tStat = 4.2445, p << 0.01) whereas opposite-side inactivation had no significant effect on the proportion of units preferring either cue-contra (tStat = 0.5689, p = 0.57) or cue-ipsi (interaction term; tStat = −1.1691, p = 0.24). In addition, both regression models indicated

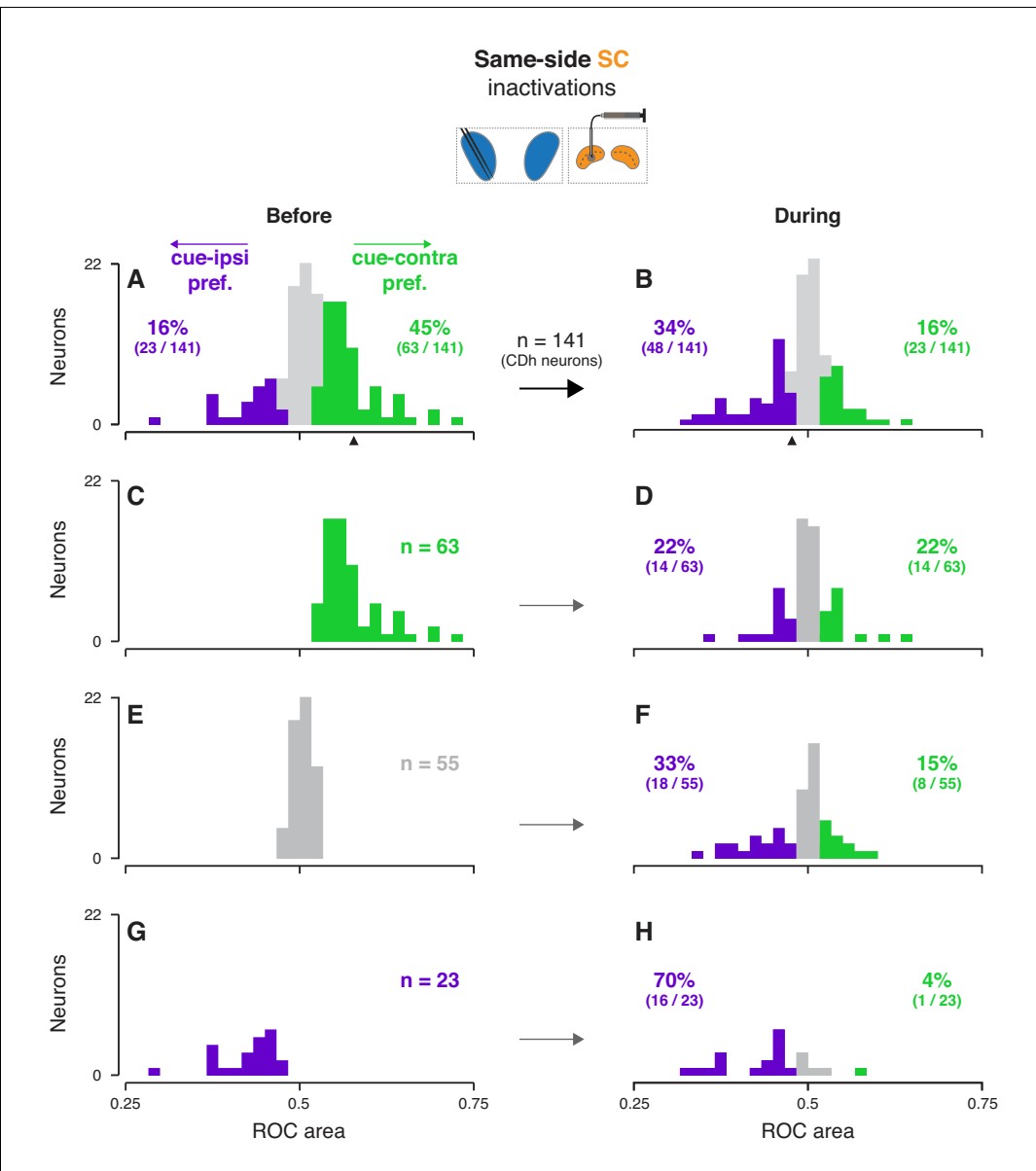

**Figure 3.** Effects of same-side SC inactivation on cue-side preferences in continuously isolated CDh neuron subpopulation. (A) Distribution of individual CDh across-task-epochs ROC areas (cue-side preferences) before same-side SC inactivation; colored bars indicate neurons with significant cue-contra (green; ROC area CI >0.5) and cue-ipsi (purple; ROC area CI <0.5) preferences and gray bars indicate no-preference (ROC area CI ⊂ 0.5); colored-text percentages (and fractions) indicate the percentage (fraction) of units with significant cue-contra (green) or cue-ipsi (purple) preferences; black triangle below axis indicates ROC area value for example unit presented in *Figure 2A–D*. (B) Distribution of ROC areas (cue-side preferences) during same-side SC inactivation; presentation as in (A). (C) Distribution of before-inactivation ROC areas (cue-side preferences) restricted to CDh neurons with significant cue-contra preferences (green text indicates number of neurons with this preference); bar heights are identical to green bars in panel (A). (D) Distribution of during-inactivation ROC areas (cue-side preferences) restricted to CDh neurons with significant before-inactivation cue-contra preferences; colored text indicates the proportion (fraction) of units with significant before-inactivation cue-contra preferences that had significant during-inactivation cue-contra preferences (green) or cue-ipsi preferences (purple). (E) Presentation as in (C) but restricted to CDh neurons with no-preference before inactivation. (F) Presentation as in (D) but restricted to CDh neurons with no-preference before inactivation. (G) Presentation as in (C) but restricted to CDh neurons with significant before-inactivation cue-ipsi preferences. (H) Presentation as in (D) but restricted to CDh neurons with significant before-inactivation cue-ipsi preferences.

The online version of this article includes the following figure supplement(s) for figure 3:

*Figure 3 continued on next page*

*Figure 3 continued*

**Figure supplement 1.** Effects of SC inactivation on CDh neuron firing rates.
**Figure supplement 2.** Effects of opposite-side SC inactivation on cue-side preferences in continuously isolated CDh neuron subpopulation.
**Figure supplement 3.** Effects of same-side SC inactivation on cue-side preferences in continuously isolated CDh neuron subpopulation (monkey P).
**Figure supplement 4.** Effects of same-side SC inactivation on cue-side preferences in continuously isolated CDh neuron subpopulation (monkey R).

no effect of task-epoch (same-side model tStat = 0.3151, p = 0.75; opposite-side model tStat = −1.3342, p = 0.18).

A difference in independent subpopulation cue-side preference distributions (ROC areas; during minus before; *Figure 4I*) illustrates that during same-side SC inactivation, the reduction of significant CDh cue-contra preferences was accompanied by an increase in non-significant cue-ipsi preferences; the reverse trend was evident for opposite-side SC inactivation (*Figure 4J*). Quantitatively, same-side SC inactivation caused the skewness of the ROC area distribution to change, significantly, from positive to negative (before: skewness = 0.12, 95% bootstrapped CI = [−0.05, 0.29]; during: skewness = −0.29, 95% CI = [−0.47,−0.11]; p0.05; bootstrap test), and caused a small but significant change in the distribution's mean (before: μ = 0.515, 95% bootstrapped CI = [0.509, 0.520]; during: μ = 0.478, 95% CI = [0.474, 0.482]; p 0.05, bootstrap test). As in our continuously isolated CDh subpopulation, the categorical shift in independent CDh subpopulation cue-side preferences caused by same-side SC inactivation was due to a redistribution of preferences, not just an overall translation of the distribution of ROC areas.

Importantly, because the SC is known to play an important role in the generation of microsaccades (*Hafed et al., 2009*), we confirmed that the redistribution of CDh cue-side preferences observed during same-side SC inactivation was independent of microsaccades (*Figure 4—figure supplement 3*).

In summary, these findings demonstrate that the modulation of CDh neuronal activity by spatial cues depends directly or indirectly on activity from the SC on the same side of the brain.

## Changes in spike-count correlations of CDh neurons

We next examined pairwise spike-count correlations to assess whether SC inactivation altered the influence of some common input signal to CDh (*Cumming and Nienborg, 2016*). We computed correlations in all combinations of task-epoch, cue-side condition, and inactivation state (before/during) among pairs of neurons in our independent subpopulations. Values were pooled across epochs and cue-sides because there were no significant differences across these conditions (bootstrap tests, smallest p = 0.21). We then fit the resulting distributions simultaneously (see Methods) and used the fits to determine whether inactivation had changed the shape of the distributions (*Figure 5*; see *Figure 5—figure supplements 1* and *2* for monkey-specific data). Fitting showed that the distribution of correlations during same-side SC inactivation (*Figure 5C*) was significantly narrower than before (*Figure 5A*; bootstrap test, p << 0.01); the width parameter of the fitted table distribution narrowed from 0.071 (95% bootstrapped CI = [0.07, 0.072]) to 0.057 (95% CI = [0.056 0.058]). A difference of density histograms (during minus before) illustrates that same-side inactivation caused a reduction in both positively and negatively correlated pairs, and an increase in weakly correlated and uncorrelated pairs (*Figure 5E*). In contrast, opposite-side SC inactivation caused no change in the width of correlation distributions (*Figure 5B and D*; bootstrap test, p = 0.88; fitted table distribution width parameter before = 0.062, 95% CI = [0.061, 0.063]; during = 0.061, 95% CI = [0.06, 0.062]), but did result in a significant increase in skewness from 0.76 (95% bootstrapped CI = [0.64, 0.97]) to 1.88 (95% CI = [1.68, 2.09]), which can be seen in the small increase of positive values in the during minus before difference of density histograms (*Figure 5F*).

Because pairs of neurons with lower firing rates tend to be less strongly correlated (*Cohen and Maunsell, 2009*; *Ecker et al., 2010*; *de la Rocha et al., 2007*; *Mitchell et al., 2009*), we compared firing rates before SC inactivation to firing rates during inactivation. We compared firing rates at the level of single continuously isolated CDh neurons (*Figure 3—figure supplement 1G,H*) and at the population level for both continuously isolated and independent subpopulations (*Figure 3—figure*

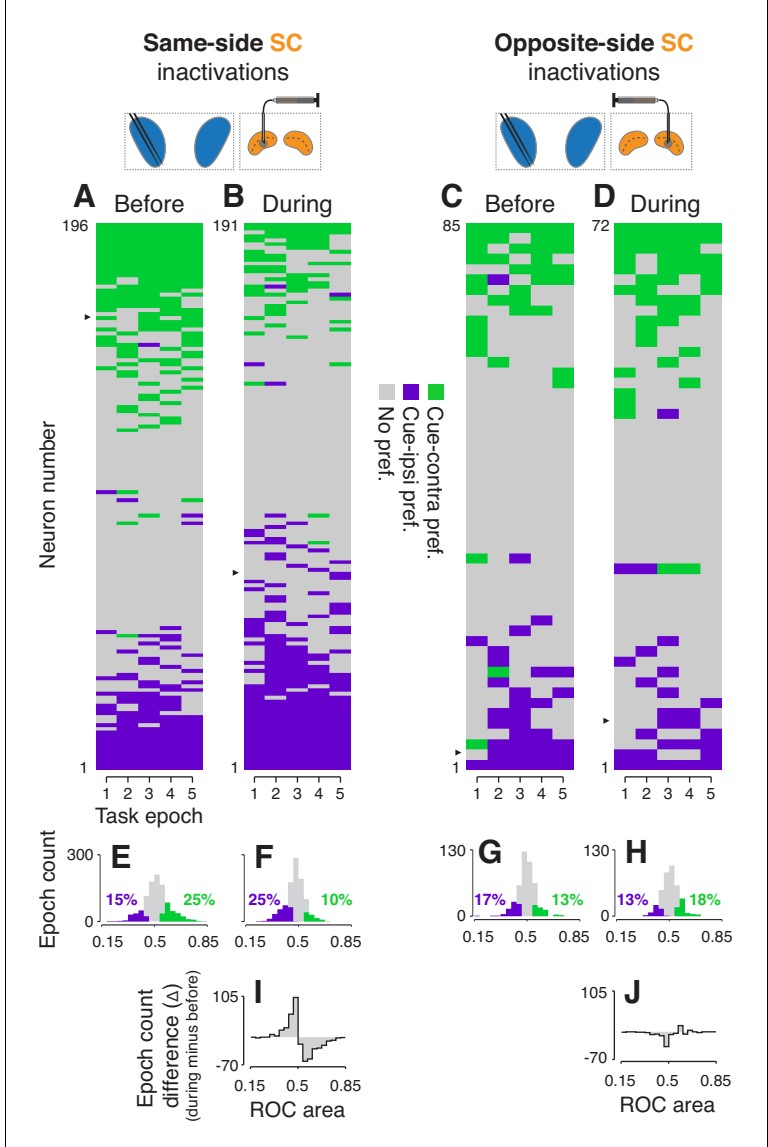

**Figure 4.** Effects of SC inactivation on cue-side preferences in independent CDh neuron subpopulations. (**A**) Cue-side preferences in each task-epoch (columns) for each CDh neuron (rows) recorded before same-side SC inactivation. Green areas indicate significant cue-contra preference, gray areas indicate no significant preference, and purple areas indicate significant cue-ipsi preference. Rows (neurons) are sorted from most cue-contra preference at top to most cue-ipsi preference at bottom. Black arrowhead at left edge indicates the example neuron shown in *Figure 2A*. (**B**) Presentation as in panel (**A**), but for neurons recorded during same-side SC inactivation. Black arrowhead at left edge indicates the example neuron shown in *Figure 2B*. (**C**) Presentation as in panel (**A**), but for neurons recorded before opposite-side SC inactivation. Black arrowhead at left edge indicates the example neuron shown in *Figure 2E*. (**D**) Presentation as in panel (**A**), but for neurons recorded during opposite-side SC inactivation. Black arrowhead at left edge indicates the example neuron shown in *Figure 2F*. (**E**) Histogram of ROC areas comparing cue-contra to cue-ipsi for all CDh neurons recorded before same-side SC inactivation, collapsed across task-epochs. Green bars indicate significant cue-contra preferences, and purple bars indicate significant cue-ipsi preferences. Colored text indicates percentages of significant cue-side preference epochs. (**F**) Presentation as in (**E**), but for during same-side SC inactivation. (**G**) Presentation as in (**E**) but for before opposite-side SC inactivation. (**H**) Presentation as in (**E**) but for during opposite-side SC inactivation. (**I**) Difference-of-histograms plot showing the change in ROC area distribution during same-side SC inactivation. Histogram values in (**E**) were subtracted from values in (**F**), ignoring cue-side preferences. (**J**) Presentation as in (**I**) but for opposite-side SC inactivation data.

The online version of this article includes the following figure supplement(s) for figure 4:

*Figure 4 continued on next page*

*Figure 4 continued*

**Figure supplement 1.** Effects of SC inactivation on cue-side preferences in independent CDh neuron subpopulations (monkey P).

**Figure supplement 2.** Effects of SC inactivation on cue-side preferences in independent CDh neuron subpopulations (monkey R).

**Figure supplement 3.** CDh cue-side preference changes during SC inactivation were unrelated to small saccadic eye movements (microsaccades).

*supplement 1I–P*). In contrast to expectations given the observed decrease in absolute correlation strength during same-side SC inactivation, individual neuron firing rates increased in 55% (78/141) of continuously isolated CDh neurons (all permutation-test 95% ROC area CIs > 0.5), decreased in 39% (55/141; all 95% CIs < 0.5), and were unchanged in 6% (8/141; all 95% CIs $\subset$ 0.5), compared to before inactivation. During opposite-side SC inactivation, individual firing rates increased in 47% (25/53; all 95% CIs > 0.5), decreased in 49% (26/53; all 95% CIs < 0.5), and were unchanged in 4% (2/53; all 95% CIs $\subset$ 0.5). At the population level, firing rates increased non-significantly during same-side

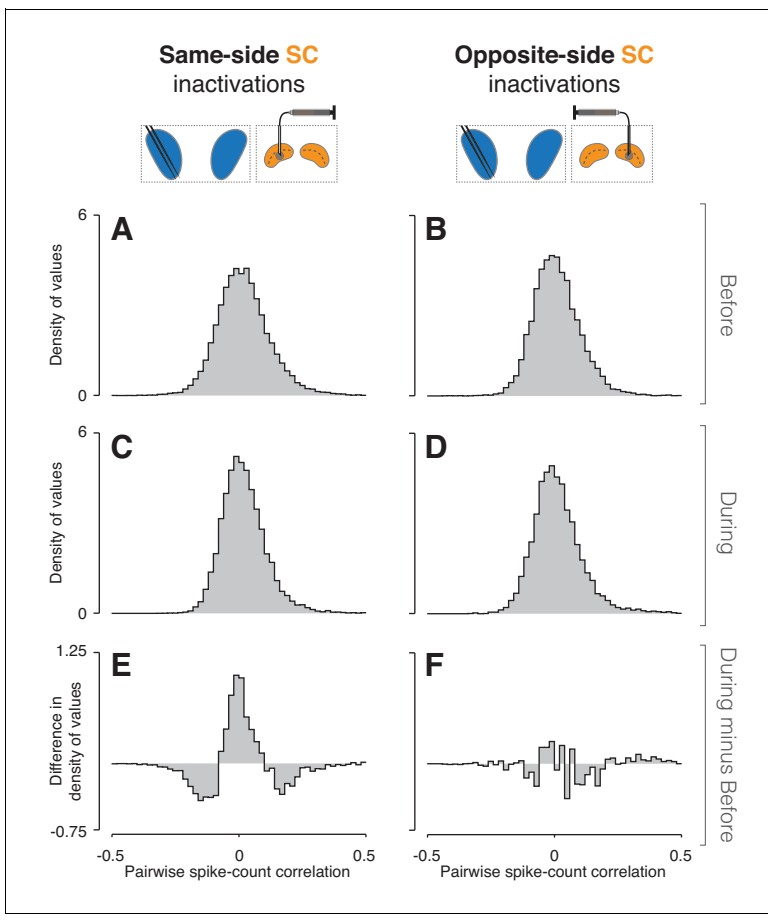

**Figure 5.** Inactivation effects on CDh pairwise neuronal correlations. (**A**) Before same-side SC inactivation density histogram of pairwise spike-count correlations pooled across epochs and cue-side presentations (n = 4280 pairs). (**B**) Presentation as in (**A**) but for before opposite-side SC inactivation (n = 2302 pairs). (**C**) Presentation as in (**A**) but for during same-side SC inactivation (n = 4042 pairs). (**D**) Presentation as in (**B**) but for during opposite-side SC inactivation (n = 1468 pairs). (**E**) Density histogram difference: during same-side SC inactivation minus before same-side SC inactivation. (**F**) Presentation as in (**E**) but for opposite-side SC inactivations.

The online version of this article includes the following figure supplement(s) for figure 5:

**Figure supplement 1.** Inactivation effects on CDh pairwise neuronal correlations (monkey P).

**Figure supplement 2.** Inactivation effects on CDh pairwise neuronal correlations (monkey R).

SC inactivation in both continuously isolated (before: $\mu_{cont}$ = 5.2 sp/s, bootstrapped 95% CI = [4.3, 6.6]; during: $\mu_{cont}$ = 5.8 sp/s, CI = [4.9, 6.9]) and independent subpopulations (before: $\mu_{ind}$ = 6.2 sp/s, CI = [5.4, 6.9]; during: $\mu_{ind}$ = 6.8 sp/s, CI = [6.0, 7.6]); the lack of a significant change in population firing rates during inactivation was evident within sessions (smallest tStat = −0.6715, smallest p = 0.5) and across sessions (smallest tStat = 0.6764; p = 0.49). Population firing rates decreased non-significantly during opposite-side SC inactivation in both continuously isolated (before: $\mu_{cont}$ = 6.0 sp/s, CI = [4.7, 7.9]; during: $\mu_{cont}$ = 5.8 sp/s, CI = [4.9, 6.9]) and independent subpopulations (before: 6.2 sp/s, CI = [5, 7.7]; during: $\mu_{ind}$ = 5.9 sp/s, CI = [4.8, 7.6]); the lack of a significant change in population firing rates during inactivation was evident within sessions (smallest tStat = −0.1487; smallest p = 0.32) and across sessions (tStat = 2.1392e-12, p = 1). These findings demonstrate that SC inactivation effects on CDh spike-count correlations were unlikely to be secondary effects of inactivation on CDh firing rates.

From these results, we conclude that same-side SC inactivation reduced the influence of one or more common inputs to CDh. Because removing this common influence results in both fewer positively and fewer negatively correlated pairs, it must ultimately have excitatory effects on some CDh neurons and inhibitory effects on others.

## Changes in classifier performance based on CDh neuronal activity

To test how SC inactivation altered information content across all simultaneously recorded CDh neurons (independent subpopulations), we examined the performance of multi-dimensional classifiers, which rely on information contained in the population-level structure of neuronal activity. Although changes in cue-side preferences (*Figure 4*) and pairwise correlations (*Figure 5*) during inactivation strongly suggest changes in population-level information encoding, some factors that affect encoding (such as covariance structure and the geometrical relationship between covariance and mean) cannot be reduced to individual or pairwise neuronal measures. Specifically, we asked how well CDh population activity encoded information about the cue-side and task-epoch, and whether SC inactivation affected that encoding. Although our analysis of cue-side preference found no consistent variation with task-epoch, previous results found that caudate neuronal activity can discriminate among task-epochs during an attention task (*Arcizet and Krauzlis, 2018*), presumably because the patterns of activity across caudate neurons convey additional information about task-epochs not detected by our ROC analysis of cue-side preference.

For each session, we trained and tested two classifiers: the first with CDh neuronal activity from before SC inactivation and the second with activity recorded during inactivation (*Figure 6*; see *Figure 6—figure supplements 1* and *2* for results from individual monkeys). Each classifier took an n-dimensional vector of neuronal activity (where n is the number of simultaneously recorded neurons before or during inactivation) from one epoch in one trial and returned a 'classifier epoch index' that uniquely identified the task-epoch and cue-side (*Figure 6A*). Classifier epoch index duplicated task-epochs 1–5 to include 1–5 cue-contra and 1–5 cue-ipsi, and was compared to the 'true epoch index' (the task-epoch and cue-side from which the data actually arose) to generate a confusion matrix (*Figure 6B*) showing how frequently each classifier epoch index was correctly identified. To help interpret the confusion matrix, we also highlighted three categories of classification outcome relevant to our specific scientific questions: (1) correct classifications, (2) cue-side misclassifications, and (3) epoch misclassifications (*Figure 6B*). After comparing the performance of several classifier variants, we found that a 'boosted decision tree' yielded the best correct-classification rates over all datasets and used this variant for subsequent analyses.

Same-side SC inactivation impaired the ability of classifiers to decode both cue-side and task-epoch information. Before same-side SC inactivation, correct-classification rates were high for all epochs on both cue-sides (*Figure 6C*), in aggregate 70% across sessions and epoch indexes (*Figure 6F*). High aggregate classifier performance resulted from largely consistent patterns of performance in individual same-side SC inactivation sessions (*Figure 6F*). During same-side SC inactivation, classification performance was poor across epoch indexes (*Figure 6D*), with correct-classification rates falling to about 47% (*Figure 6G*).

To highlight the effects of SC inactivation on classifier performance, we examined a difference of confusion matrices (during minus before; *Figure 6E*). Cue-contra epoch indexes were slightly more strongly affected than cue-ipsi, with correct-classification rates falling by 26% for cue-contra and 21% for cue-ipsi (*Figure 6H*). Misclassification errors induced by same-side SC inactivation were

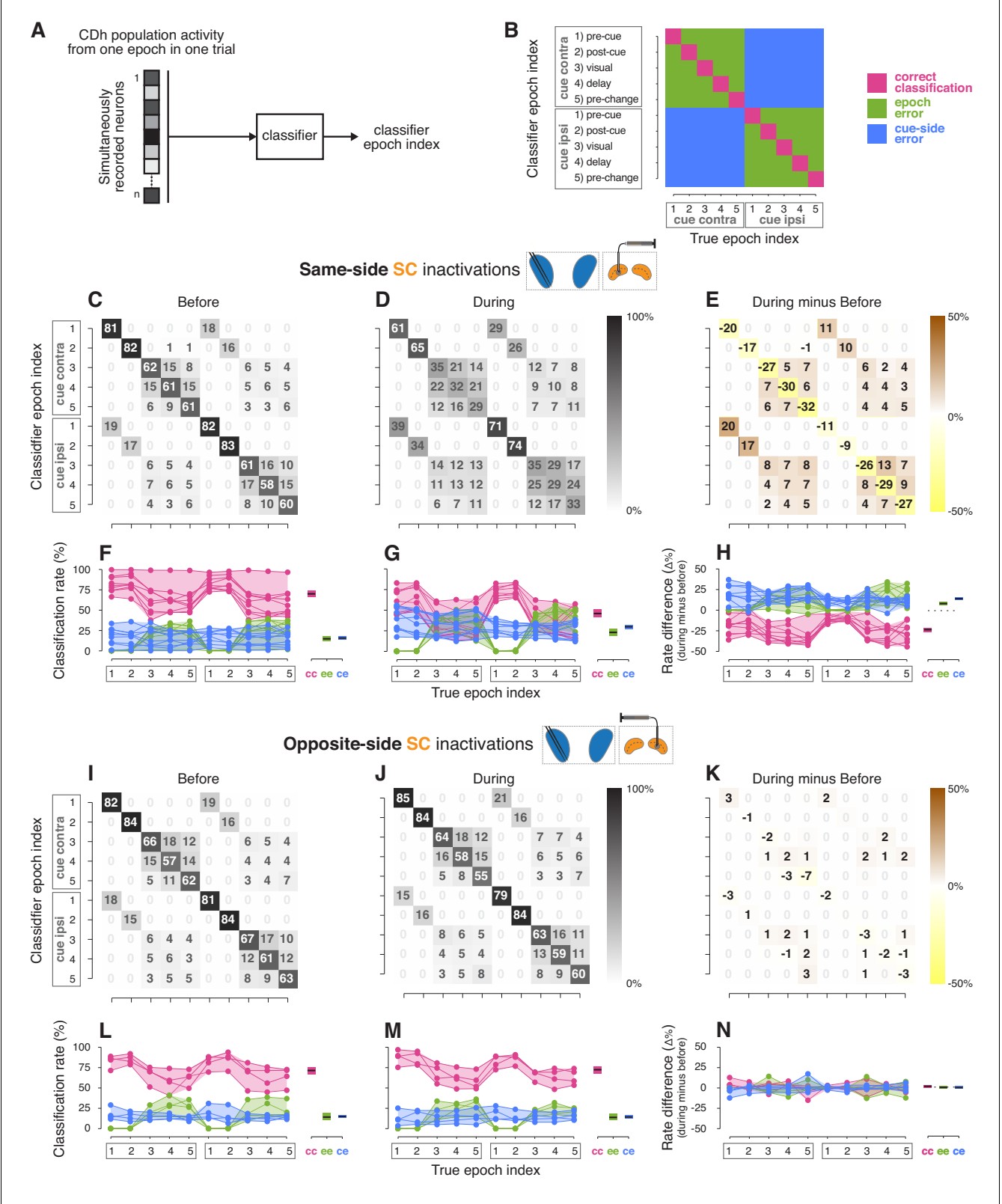

**Figure 6.** Classifier analyses. (**A**) A boosted decision tree classifier took n-dimensional vectors (where n is the number of simultaneously recorded neurons before or during SC inactivation) of CDh neuronal activity from individual epochs in single trials and returned a 'classifier epoch index' indicating the classifier's guess regarding both which task-epoch and which cue-side condition the activity came from. (**B**) A confusion matrix is traditionally used to show how frequently each possible classifier epoch index (vertical axis) is applied for each true epoch index. Here we have

*Figure 6 continued on next page*

*Figure 6 continued*

additionally color-coded a confusion matrix to highlight three categories of classification outcome: (1) correct classifications (magenta), (2) cue-side misclassifications (blue), and (3) epoch misclassifications (green). (**C**) Across sessions aggregated cross-validated confusion matrix before same-side SC inactivation. Cell shading starts at white for 0% with larger values grading darker up to black at 100%. For each 'true epoch index' column (for example, 'pre-cue cue-contra' in column 1), the numeral in each row is the (rounded) percentage of times classifiers assigned that row's class label to all inputs of that column's input (true) class. For example, the number 81 in the 1st row and 1st column indicates that, across classifiers/sessions, when classifiers were given 'pre-cue cue-contra' input vectors, 81% of those vectors were labelled as 'pre-cue cue-contra'; the number 19 in column 1, row six indicates that the remaining 19% of the time, classifiers erroneously labelled 'pre-cue cue- contra' activity vectors as 'pre-cue cue-ipsi'. (**D**) Presentation as in (**C**) but for during same-side SC inactivation. (**E**) Difference of aggregated confusion matrixes (during minus before). Cell color mapping starts at saturated yellow for −50% with larger values grading to white at 0%, followed by increasing positive values grading up to saturated brown at +50%. (**F**) Left: per-session breakdown of classifier performance with classification rates divided into outcome categories shown in panel (**B**). Connected dots are from a single session. For each true epoch index on the horizontal axis, colored dots show the percentage of times the single-session classifier epoch index fell into one of the three classification categories defined in (**B**) (correct classification, epoch misclassification, cue-side misclassification). Right: per-category breakdown of mean classification rates across sessions and true epoch indexes; cc: correct classifications, ee: epoch errors, ce: cue-side errors. Shaded areas indicate 95% bootstrapped confidence intervals on mean from individual session classification rates. (**G**) Presentation as in panel (**F**), but for during same-side SC inactivations. (**H**) Left: per-session differences in classification categories (during minus before). Right: per-category breakdown of difference in mean classification rates across sessions and true epoch indexes. Dotted line marks 0. (**I–N**) Presentation as in panels (**C–H**) but for opposite-side SC inactivation sessions.

The online version of this article includes the following figure supplement(s) for figure 6:

**Figure supplement 1.** Classifier analyses (monkey P).

**Figure supplement 2.** Classifier analyses (monkey R).

more frequently cue-side errors, which rose by 16%, than epoch errors, which rose by 9% (*Figure 6H*). To statistically test changes in classification rates caused by inactivation, we used logistic regression on rates with session, classification-outcome category, and true epoch index as predictors. This analysis confirmed that correct-classification rates were significantly poorer during same-side SC inactivation compared to before (tStat = −2.3893, p < 0.02), and that cue-side misclassification rates were more pronounced than epoch misclassification rates (interaction term: tStat = 2.2681, p < 0.03).

Opposite-side SC inactivation had essentially no effect on classification performance. Before opposite-side SC inactivation (*Figure 6I*), the aggregate correct-classification rate was 71% (*Figure 6L*) and rose to 72% during inactivation (*Figure 6J,M*). Again, these patterns of performance were consistent across sessions (*Figure 6L–M*). Logistic regression applied to data from these opposite-side SC inactivation sessions revealed no significant effect of inactivation (tStat = −0.54726, p = 0.5842), and no significant interactions (tStat range: −0.05–0.075, all p > 0.91).

From these classifier-based analyses, we conclude that same-side SC inactivation primarily disrupts CDh population encoding of the task-relevant spatial location, and also interferes with non-spatial encoding of task-epochs.

## Discussion

Our results establish that the pattern of neuronal activity observed during attention-task performance in the anterior portion or the CDh relies on output from the superior colliculus (SC), either directly or indirectly, and provide a possible basis for the reliable, performance-altering effects of SC inactivation during attention tasks. Inactivation of SC on the 'same side' of the brain as recorded CDh neurons clearly and consistently disrupted attention-related modulation in CDh whereas inactivation of 'opposite-side' SC did not. Specifically, SC inactivation redistributed the cue-side preferences of individual CDh neurons, decreasing the prevalence of cue-contra preferences and increasing cue-ipsi preferences. Inactivation also disrupted a non-spatial component of CDh activity, impairing the ability of a classifier to correctly decode task-epoch from populations of CDh neurons. Inactivation of the same-side SC also narrowed the distribution of spike-count correlations between pairs of CDh neurons, consistent with the interpretation that the SC normally provides a shared input to CDh neurons that was reduced during inactivation. These results provide causal evidence that a novel interaction between the SC and basal ganglia is part of the control system for covert visual attention.

## Could CDh neuronal changes during SC inactivation be due to nonspecific effects?

Because SC inactivation causes major deficits in the performance of spatial attention tasks, it seems possible that the changes we observed in CDh neurons might be a nonspecific result of the behavioral deficits that accompany inactivation, perhaps due to changes in overall motivation or arousal. However, the results from our control experiments in which we inactivated the SC on the side opposite to our caudate recordings indicate that this is unlikely. Opposite-side SC inactivation reduced cue-ipsi hit rates but did not affect the cue-ipsi preferences of CDh neurons. If the changes in CDh cue-side preference simply followed the behavior deficits, then the cue-side preferences should have changed after both same- and opposite-side SC inactivation. Furthermore, our classifier analyses showed that same-side SC inactivation decreased the amount of available cue-side information in CDh populations for both cue-contra and cue-ipsi conditions, whereas opposite-side inactivation left cue-side information intact for both cue-contra and cue-ipsi. These results show that CDh cue-side preferences changed only after same-side SC inactivation, consistent with the anatomical circuits that connect the SC to CDh, which are strongly ipsilaterally biased and discussed in more detail below.

Although this dissociation of cue-side effects between behavior and neurons may be surprising, previous work from our lab has shown that SC inactivation can impair attention-task behavior while attention-related modulation of the attended visual feature in sensory cortex remains intact (*Zénon and Krauzlis, 2012*). This finding demonstrates that altered attention-task behavior does not guarantee altered attention-related modulation of neuronal activity, consistent with our current results.

Because the SC is one of many brain areas that have been implicated in the control of visual spatial attention, a natural question is whether the effects we have observed are specific to causal perturbation of SC. It is possible that changes in CDh neuronal activity will result from causal perturbations of other structures (e.g. frontal eye fields; FEF) as long as they produce sufficiently strong network-wide changes in activity. However, the SC is an especially useful structure to target because focal SC inactivation causes large and easily reproducible spatially specific deficits in attention-task performance. Indeed, previous work has found that the effects of FEF inactivation on attention-task behavior are smaller and less reliable than the effects of SC inactivations (*Bollimunta et al., 2018*).

## A reversal of the classic subcortical hierarchy

Our demonstration that SC activity influences CDh is a reversal of the typically depicted subcortical hierarchy in which information flows from cortex through the basal ganglia to the SC (e.g. *Hikosaka et al., 2014*). This adds an ascending stream of interaction from SC to CDh to the already well-established circuits by which CDh output influences SC (*Yasuda and Hikosaka, 2015*). According to the classic circuit diagram, caudate output, acting through the external segment of the globus pallidus (GPe) and substantia nigra pars reticulata (SNr), modulates the excitability of SC neurons by removing or adding inhibition, thereby affecting saccade probability (*Hikosaka et al., 2006*). This mechanism can help primates orient towards visual stimuli associated with large rewards and away from those associated with small rewards (*Amita and Hikosaka, 2019*; *Yamamoto et al., 2013*).

Our results complement this picture by demonstrating the importance of signals arising from the SC and sent back directly or indirectly to the basal ganglia. There are two prominent routes by which neurons in the intermediate and deep layers of SC – which our inactivations targeted – might more directly affect CDh neuronal activity: (1) through the substantia nigra pars compacta (SNc), or (2) through the centromedian-parafascicular complex (CM-Pf) of the thalamus (*Krauzlis et al., 2013*). Previous experimental work in monkeys has indicated both of these pathways from SC to CDh might be involved in the control of spatial attention.

With regard to the first route, dopaminergic (DA) neurons in SNc can be activated by SC and broadcast signals widely throughout the striatum that are necessary for normal processing of visual information. In monkeys with V1 lesions, responses of SNc DA neurons to reward-predicting visual stimuli are virtually eliminated by SC inactivation (*Takakuwa et al., 2017*), demonstrating that primate SC can excite SNc DA neurons. When SNc DA signals to caudate are cut off by unilaterally infusing MPTP into caudate, the result is contralateral visual hemineglect (*Miyashita et al., 1995*).

Monkeys in this state are still able to make contraversive saccades when presented with single targets but fail to do so under free viewing conditions or when presented with a pair of lateralized saccade choice targets. These results are consistent with the contralateral attention deficits caused by SC inactivation being partly mediated by a circuit through SNc to caudate.

The second route, from SC through the CM-Pf complex to CDh, has been implicated in the control of attention in both monkeys and humans. Neurons in the intermediate and deep layers of SC send a predominantly ipsilateral projection to the CM-Pf (*Harting et al., 1980*; *Partlow et al., 1977*) which, in turn, sends an exclusively ipsilateral projection to the striatum (*Matsumoto et al., 2001*; *Nakano et al., 1990*). Unilateral muscimol inactivation of CM-Pf in monkeys removed the reaction time (RT) benefit conferred by a spatial cue in an attention task, but only for the contralateral hemifield not the ipsilateral field (*Minamimoto and Kimura, 2002*). In humans performing a spatial attention task, fMRI data indicated that the CM-Pf was consistently activated by the 'attentional shifts' required in that task (*Hulme et al., 2010*). These data are consistent with the SC transmitting signals related to attentional selection of contralateral visual stimuli to caudate through CM-Pf.

In addition to the relatively direct anatomical routes described above, SC inactivation could also affect CDh neurons by other circuits or through less direct network-level effects. Inactivation of the SC is likely to influence neuronal activity in a number of brain areas, and the long-lasting effects of muscimol increase the likelihood that neuronal networks spanning multiple areas will adapt their activity in response to the SC's silencing by inactivation. While our results establish the causal dependence of CDh neuronal activity on intact SC output, they do not reveal the specific circuits or networks responsible. Experiments to determine the mechanism(s) by which SC can influence CDh will be facilitated by the development of primate optogenetic tools, particularly those suited for subcortical applications (*Cushnie et al., 2020*). Specifically, such tools will allow manipulation of neuronal activity with fine spatial and temporal precision, making it possible to target specific area-to-area projections on the timescale of milliseconds. In the present context, a strong test of the proposition that circuits from the SC through SNc or CM-Pf mediate the effects of SC inactivation on CDh neurons could be achieved by recording from CDh neurons during attention-task performance while transiently silencing signals from the SC through one or both of those putative intermediate nodes; if such transient silencing of one or both pathways resulted in concurrent shifts of CDh cue-side preferences it would suggest a more direct mechanism, but if longer-duration silencing or silencing of additional SC output pathways was required to influence CDh, a network-level effect would seem more likely.

Irrespective of mechanism, our finding that same-side SC inactivation strongly alters CDh neuron cue-side preferences establishes the importance of recurrent interactions between the SC and basal ganglia during covert visual attention tasks. This interaction includes modulation of neuronal excitability in SC by basal ganglia output and shaping of spatial selectivity in caudate neurons by SC. Our proposal for the functional role of this 'subcortical loop' differs from previous conceptions (*McHaffie et al., 2005*; *Redgrave, 2010*; *Redgrave and Gurney, 2006*) in the primary emphasis we place on a visual selection mechanism. An abundance of evidence has accumulated for the SC's causal participation in visual selection – SC activity helps determine which visual information is used to guide perceptual reports (*Cavanaugh and Wurtz, 2004*; *Herman et al., 2018*; *Lovejoy and Krauzlis, 2010*; *Müller et al., 2005*; *Zénon and Krauzlis, 2012*). Attention is impaired when SC is inactivated, or when the nodes allowing SC activity to reach caudate are lesioned. Together with the CDh neuronal correlates of SC inactivation we observed, we consider this strong evidence that disruption of the recurrent interactions between SC and basal ganglia are a major reason why SC inactivation causes deficits in covert visual attention.

## Task-state representation

Our classifier results are compatible with striatal circuits representing 'belief states' that summarize moment-by-moment environmental conditions. As an example, belief states can support the identification of conditions that predict upcoming sensory events (*Starkweather et al., 2018*). A number of studies have identified features of striatal neuron activity that would support belief state encoding, namely phasic activation by a variety of sensory and motor events at a range of delays. The activation of unique subsets of striatal neurons at distinct times has been variously interpreted to mean this activity represents action sequences (*Jog et al., 1999*; *Kermadi and Joseph, 1995*; *Miyachi et al., 2002*; *Sheng et al., 2019*), working-memory signals (*Akhlaghpour et al., 2016*),

accumulated evidence (*Ding and Gold, 2010*), uncertainty about object-reward associations (*White et al., 2019*), stimulus or reward expectation (*Hikosaka et al., 1989*), or action value (*Lau and Glimcher, 2008*; *Lee et al., 2015*; *Samejima et al., 2005*; *Seo et al., 2012*). We consider the heterogenous joint representation of multiple task- and internal-state variables displayed by striatal neurons across experimental conditions as most consistent with striatal population encoding of 'belief state' (*Daw et al., 2006*; *Rao, 2010*). Belief states provide an elegant way to apply reinforcement learning (RL) methods to the problem of learning environmental state-contingent action values in the face of uncertainty about environmental state, which aligns with inherently stochastic sensory representations found in the brain (*Gershman and Uchida, 2019*).

Striatal encoding of belief state is compatible with our classifier results and offers a novel interpretation of changes in attention-task behavior resulting from SC inactivation. If striatal neurons represent belief state, it should be possible to accurately decode environmental conditions from the activity of those neurons, as our classifier results demonstrated. The defining features of our attention-task space (cue-side and task-epoch) amount to the task's 'environmental conditions', which we found can be decoded from CDh neuronal activity with high accuracy (~70%) before SC inactivation (*Figure 5*; *Arcizet and Krauzlis, 2018*). The impaired decoding of task-state observed during same-side SC inactivation then implies a disorganized underlying belief state representation. If SC inactivation degrades belief state representation, the behavioral effects of SC inactivation during attention tasks amount to errors in estimating the value of responding or withholding responses for task events. In contrast to standard views of attention, which focus on the quality of sensory representations, this interpretation suggests that some of the behavioral and neurophysiological correlates of attention can be understood as consequences of mechanisms for learning the value of actions or the value of state-contingent action values (sensory-motor associations). From this perspective, SC inactivation reduces hit rate by reducing the subjective value of responding to cued changes and increases false-alarm rate by inflating the subjective value of responding to foil changes.

Our assertion that some correlates of attention arise from learning mechanisms is bolstered by the striking correspondence between subcortical structures necessary for attention and those implicated in learning. Specifically, anatomical routes that have been identified as vital for striatal expression of associations between sensory events and appetitive rewards are precisely those that could allow SC output to influence CDh neurons. In monkeys that have learned to associate a 'conditioned stimulus' (CS) with either an appetitive or an aversive stimulus, tonically active neurons (TANs) in the striatum exhibit stereotypical 'pauses' in spiking for visual or auditory CS presentation (*Aosaki et al., 1994b*), but not for unpaired stimuli (for which no association has been learned). These learning-induced responses of striatal TANs require both intact SNc DA projections to striatum (*Aosaki et al., 1994a*), and normal signaling from CM-Pf (*Matsumoto et al., 2001*). Striatal TANs are thought to be large aspiny cholinergic interneurons which can bidirectionally regulate the influence of cortical input to MSNs pre- and post-synaptically (*Ding et al., 2010*). These previous results show that visual stimulus-related signals flowing through SNc and CM-Pf to the striatum have the potential to alter the sensitivity of inputs to striatal MSNs (such as those we recorded from) by affecting the activity of striatal TANs. The same routes through SNc and CM-Pf that likely allow SC to influence attention-related activity in caudate are also necessary for expression of learning-related activity changes in striatum.

## Conclusions

We have provided causal evidence that attention-related modulation in caudate neurons depends on output from the superior colliculus. Intact signals from the SC are also necessary for CDh population representation of task-state variables. Although the specific circuit or network mechanisms responsible for these effects remains to be identified, our results offer a novel interpretation of attention-task deficits during SC inactivation as the result of altered processing of stimulus relevance in the basal ganglia and demonstrate the general importance of the basal ganglia for the control of spatial attention.

## Materials and methods

### Animals

Data were collected from two adult male rhesus macaques (P and R; *Macaca Mulatta*) weighing 10–16 kg. All experimental protocols were approved by the National Eye Institute Animal Care and Use Committee, and all procedures were performed in accordance with the United States Public Health Service policy on the humane care and use of laboratory animals.

### Tasks and stimuli

The details of our apparatus, task, and stimuli have been described in detail previously (*Arcizet and Krauzlis, 2018*), but are partially reproduced here for clarity. Monkeys were seated in primate chairs (Crist Instrument Co., Hagerstown, MD, United States) with head fixed, inside a darkened booth. Animals were positioned with eyes 48 cm from an LCD display with a refresh-rate of 100 Hz (VIEW-Pixx; VPixx Technologies, Saint-Bruno, QC, Canada), and experiments were orchestrated with a modified version of PLDAPS (*Eastman and Huk, 2012*), running on a 'MacPro5,1' (Apple Inc, Cupertino, CA). Eye position was monitored using an EyeLink 1000 infrared eye-tracking system (SR Research, Ottawa, Ontario, Canada), and manual responses were collected with a single-axis joystick (CH Products, model HFX-10) mounted to the primate chair and oriented to allow vertical movement.

Monkeys initiated each trial by pressing down on a joystick which triggered the appearance of a central fixation square. Central gaze fixation within a 2–3° window was required for the entire duration of the trial; exiting the window caused the trial to be aborted and repeated. After an initial 250 ms of maintained fixation, a cue-ring (inner radius 3.75°, outer radius 4°) was presented for 200 ms at an eccentricity of 10° (*Figure 1B*). Two motion-dot patches (3° radius) were presented 500 ms after the cue-ring was extinguished; the 'cued' patch was centered on the former location of the cue-ring, and the 'foil' patch was presented at an equally eccentric opposing location (180° of elevation away). The location of the spatial cue was constant for a block of 68 trials and then switched to the opposite location. A motion-direction change was possible 1000 ms – 4000 ms following stimulus onset (uniform distribution), and stimuli were extinguished 1000 ms after the change (i.e., maximum stimulus duration was 5000 ms). If the cued stimulus changed, the monkey was required to release the joystick in a time window 200 ms - 800 ms after the change; if the foil stimulus changed the monkey was required to maintain joystick press for an additional 1000 ms. If the monkey released the joystick for a cued change ('hit') or maintained joystick press for a foil change ('correct reject'), a liquid reward was delivered 1000 ms after the change; no reward was delivered for failing to release the joystick for a cued change ('miss') or releasing the joystick for a foil change ('false alarm'). In each trial, either the cued stimulus changed, or the foil stimulus changed, but not both.

Because of the idiosyncratic and heterogeneous quality of the stimulus-placement preferences of caudate neurons we observed previously (*Arcizet and Krauzlis, 2018*), we chose stimulus locations to ensure behavioral effects of superior colliculus (SC) inactivation. The two diametrically opposed stimulus locations were chosen at the start of each session on the basis of the planned inactivation-deficit area so that, during SC inactivation, one of the two stimuli would fall inside the deficit area and one outside.

The direction of motion of dots in each stimulus patch was drawn from gaussian distributions with standard deviation $\sigma = 16°$. The magnitude of the motion-direction change was adjusted from session-to-session to maintain relatively constant performance and was generally kept in a range of 13° - 22°. The mean of the distribution for the cued stimulus patch and the foil stimulus patch varied from day to day but always differed by 90°. Each dot was six pixels in diameter, moved at a speed of 15 degrees per second, had a lifetime of 100 ms (10 frames), and overall dot density was 26 per square degree per second.

The delayed visually guided saccade task used to assess the effects of muscimol injection into SC has been described previously (*Zénon and Krauzlis, 2012*).

### Neuronal recordings and inactivation

At the start of each session, two 32-contact 'v-probes' (200 μm spacing between contacts in a single-column geometry; Plexon Inc, Dallas, TX) were advanced into the left CDh, and one injection

canula was advanced into cortex overlying SC; both probes and injection canula were controlled with a micromanipulator (NAN Instruments LTD, Nof Hagalil, Israel). CDh recording sites ranged from AC+2 to AC+8 (*Figure 1—figure supplement 3*), where AC refers to the anterior-posterior location of the anterior commissure, located at approximately AP20 relative to ear-bar zero with structural MRI images. We localized recording contacts to the CDh based on position information from structural MRI and on the basis of the low background activity observed on the contacts. In monkey P, we used three recording 'tracks' across experiments (referred to as tracks 1, 2, and 3; *Figure 1—figure supplement 3*). Track one was located at AC+5 (5 mm anterior to the anterior commissure: 'AC'), track two at AC+6, and track three at AC+8. In all five same-side inactivations, we used tracks 2 and 3; in one opposite-side inactivation we used tracks 2 and 3 and in the other we used track 1 and 3. In monkey R, we used two recording tracks across experiments (tracks 1 and 2; *Figure 1—figure supplement 3*). Track one was located at AC+2 and track two at AC+4. In all six experiments in this monkey (four same-side, two opposite-side inactivations), we used track 1 and track 2.

CDh neuronal data were collected over 6–8 blocks (408–544 trials) of attention-task performance before SC inactivation. Once before-inactivation data collection was complete, the injection canula was advanced to an estimated depth of 2.5 mm below the dorsal surface of the SC, at which depth 0.5 µL of 5 µg/µL muscimol (a GABA$_A$ agonist) was injected. Following muscimol injection, peak velocities of visually guided saccades to a variety of locations in the visual field were measured and compared to those from previous (non-inactivation) sessions to confirm the presence and spatial extent of a deficit in the contralateral visual field. In 9/13 'same-side' SC inactivation sessions (five in monkey P and four in monkey R), muscimol was injected into left SC, and in the remaining 4/13 'opposite-side' inactivations (two in monkey P and two in monkey R), the injection was into right SC. Once a saccade deficit was confirmed, during-inactivation CDh neuronal data were collected over 8–14 blocks (544–952 trials) of the attention task. Following collection of CDh neuronal data in the attention task during-inactivation, visually guided saccades were again used to map the spatial extent of the inactivation-deficit area.

To quantify the stability of each animal's behavior in each session (independent of SC inactivation effects), we subdivided the 'before' and 'during' portions of each session into two halves and computed hit rate (separately for each cue-side condition) in each half. This yielded eight hit rates for each session ([1$^{st}$ half of before, 2$^{nd}$ half of before, 1$^{st}$ half of during, 2$^{nd}$ half of during] X two cue-side conditions); we then compared hits in the 1$^{st}$ half to the 2$^{nd}$ half separately for both cue-side condition and for before/during using $\chi^2$-proportion tests (four tests per session). Of the 26 'before' portions examined this way (13 sessions X two cue-side conditions), we identified six with significantly different hit rates in the 1$^{st}$ half compared to the 2$^{nd}$ half (all p 0.05; largest $\chi^2 = 15.6564$); of the 26 'during' portions, three showed significantly different rates (all $p < 0.05$; largest $\chi^2 = 6.2013$). These nine cases were roughly equally likely to occur in same side inactivations (6/36 portions; 17%) and opposite-side inactivations (3/16 portions; 19%). However, in all nine cases, hit rate was *larger* in the 2$^{nd}$ half compared to the 1$^{st}$ half (of either before *or* during portion). Also, the three cases in which performance improved significantly during SC inactivation were conditions in which the spatial cue was presented outside the inactivation-deficit area. This suggests that any effects observed during SC inactivation are unlikely to be attributable to a lack of motivation or engagement with the task on the part of the animal.

## Spike sorting and unit selection

Raw voltage signals from each v-probe contact were digitized (40 kHz sample-rate), high-pass filtered and stored with an 'Omniplex D' system (Plexon Inc, Dallas, TX). These 'continuous spike channel' data were analyzed offline with Kilosort (*Pachitariu et al., 2016*), including manual curation, to identify putative single-neuron waveforms. Our intention was to identify and analyze spike data from caudate 'medium spiny projection neurons' (MSNs) also called 'phasically active neurons' (PANs). To identify MSNs, we followed previously established methods based on each putative neuron's waveform characteristics, inter-spike-interval (ISI) distribution, and firing-rate distribution (*Berke, 2008*; *Berke et al., 2004*). First, we excluded putative neurons with waveforms that did not have the pattern of large initial negative deflection ('valley') followed by a smaller positivity ('peak') that is typical of extracellular action potentials. Second, we required putative MSN waveforms to have a valley width >100 µs, peak width >350 µs, and average firing rate <20 spikes per second. Third, we

excluded neurons with an initial gap in their ISI distributions, which is typical of striatal 'tonically active neurons' (TANs). Following these criteria, of 483 clearly isolatable waveforms, we categorized 281 as MSNs (171 from monkey P and 110 from monkey R) which were used for all data analyses.

Because we used acute recording techniques, in which the relative positions of neural tissue and recording contacts may drift over the course of a recording session, we took additional steps to identify which CDh neurons were likely to have been recorded with stable isolation throughout each session. Building on previously established methods for identifying individual neurons recorded across successive days on the same channel of multi-channel 'utah' arrays (*Eleryan et al., 2014*; *Fraser and Schwartz, 2012*), we identified a 'continuously isolated' subpopulation of 194/281 CDh neurons that were likely to have been recorded stably throughout the 'before inactivation' and 'during inactivation' phases of each session (141 from same-side SC inactivation sessions, 87 from monkey P and 54 from monkey R; 53 from opposite-side, 40 from monkey P, 13 from monkey R). Our approach relied on the assumption that neurons were stably isolated throughout the before-inactivation phase. This assumption allowed us to establish stable isolation criteria by splitting data collected before inactivation into two halves and comparing spike metrics from one half to the other. We then compared spike metrics for each neuron before and during inactivation and considered those that met our stable isolation criteria 'continuously isolated' throughout the session. More specifically, we used data collected before inactivation to identify 'true positive' (same neuron) and 'true negative' (different neuron) example cases. All 483 clearly isolatable waveforms were considered true positives: we built true positive distributions of several spike metrics by comparing those metrics for each waveform computed on data from the 1st half of before inactivation to those metrics computed on data from the 2nd half of before inactivation. We built true-negative distributions by identifying 185 instances in which two different waveforms appeared on the same electrode contact, and comparing spike metrics computed on data for one waveform from the 1st half of before inactivation to metrics computed on data for the other waveform from the 2nd half of before inactivation. The five spike metrics we considered were: (1) average waveform 'shape', (2) waveform amplitude (peak-to-trough), (3) waveform power, (4) time from peak-to-trough, and (5) ISI distribution. We compared two average waveform 'shapes' by computing their Pearson correlation (each average waveform comprised 2 ms of data sampled at 40 kHz or 81 samples). We compared all other metrics by computing the Kolmogorov-Smirnov distance between distributions of metric values (for example, between distributions of time from peak-to-trough values). These steps resulted in five spike-metric values for each true positive and true negative which we then used to construct a 5-fold cross-validated regularized logistic regression model (the 'continuous isolation model'; regularization was used to contend with correlations amongst spike-metric predictors). We then applied the continuous isolation model to comparisons of spike metrics before inactivation to those metrics during inactivation in all 263/281 neurons recorded throughout before and during phases of each experiment and labelled those with a model value greater than 0.95 (74%; 194/263) as 'continuously isolated'. As an additional validation step, we applied the continuous isolation model to the (artificially split) 1st and 2nd halves of single electrode recordings of putative MSNs from the caudate nucleus of the same monkeys collected previously (*Arcizet and Krauzlis, 2018*), and found that it identified 97% (220/227) as continuously isolated, suggesting an appropriately conservative model.

## Data analysis

All data analyses were performed in MATLAB (MathWorks, Natick, MA).

Inactivation-deficit maps were generated by comparing the peak velocity of visually guided saccades during SC inactivation to baseline peak velocities (*Figure 1—figure supplement 2*). Separately for each monkey, a baseline saccade parameters dataset from more than 20000 saccades was collected over months prior to experimental data collection. To estimate the area of visual space affected by muscimol injection in each inactivation plus recording session, the peak velocity of each visually guided saccade collected during inactivation was compared to baseline peak velocity. Specifically, for each during-inactivation saccade vector we identified a set of baseline saccade vectors with endpoints falling within 0.5°, and examined this set's distribution of peak velocities; the during-inactivation saccade was scored as significantly affected by inactivation if its peak velocity fell in the bottom 5% of the baseline set's distribution. To define a session's inactivation-deficit map we used constrained nonlinear optimization to find a contour separating affected from unaffected saccade endpoints under the constraint that the contour's maximum curvature was smaller than the minimum

distance between endpoints (*Kano and Fujioka, 2018*). This constraint was used to ensure that deficit maps did not artificially inflate the spatial precision with which affected and unaffected areas could be identified.

To minimize the impact of any slow fluctuations over the course of a recording session (*Bondy et al., 2018*), each neuron's spike-count data were z-scored separately in each non-overlapping successive 'block-pair'. For example, blocks 1 and 2, comprising 136 (2 × 68) trials was considered a block-pair and included 68 trials with the cue-contralateral to recorded CDh neurons (cue-contra) and 68 trials with cue-ipsi. To keep block-pairs non-overlapping, blocks 3 and 4 made up the next block-pair (each block contributed to only one block-pair); block-pairs also exclusively consisted of before inactivation or during inactivation data, never both. Separately for each neuron and for each of several window-durations, the mean and standard deviation of spike-count (or spike-rate) values was estimated in each block-pair; z-score was computed from spike-count/rate by subtracting the mean and dividing by the standard deviation. Z-scored spike counts were used in computing cue-side preferences and for classifier analyses.

To specifically quantify the effects of SC inactivation on cue-related modulation, we focused our analyses of neuronal activity on times before the motion-direction change. We previously found that many caudate MSNs display activity related to joystick release or stimulus-change-related activity modulated by joystick release (*Arcizet and Krauzlis, 2018*). Because SC inactivation systematically altered the probability of reporting the motion-direction change, alterations in CDh activity after the stimulus change could be secondary effects caused by changes in motor behavior, so these later epochs were excluded from analysis.

Individual CDh neuron cue-side preferences were determined by calculating the area under a receiver operating characteristic (ROC) curve (*Green and Swets, 1966*). Cue-side preference in a given task-epoch was determined by computing ROC area comparing the distribution of z-scored spike counts in that task-epoch for the cue-ipsi condition (labelled the 'noise' distribution), to the distribution for the cue-contra condition (labelled the 'signal' distribution). Accordingly, an ROC area greater than 0.5 indicated a cue-contra preference and less than 0.5 a cue-ipsi preference. A cue-side preference was considered significant if the 95% confidence interval (CI) on ROC area did not contain ($\not\subset$) 0.5. The 95% CI was a 'permutation test' CI determined by shuffling the 'signal' and 'noise' labels on the distribution values and recomputing ROC area 10000 times to build a distribution and identifying the $2.5^{th}$ and $97.5^{th}$ percentiles of that distribution as the lower and upper limits of the CI, respectively. To compute an across epochs cue-side preference, the mean z-scored spike count across cue-side conditions was first subtracted from each epoch's counts, then cue-contra and cue-ipsi counts were separately pooled across task-epochs and ROC area and associated 95% CI was computed as described above.

To determine the effect of microsaccades on CDh cue-side preferences (ROC areas), those preferences were recomputed from a subset of trials without microsaccades. Microsaccades were algorithmically detected in 2D velocity space following established methods (*Engbert and Kliegl, 2003*; *Engbert and Mergenthaler, 2006*). We then defined a 'microsaccade counting window' for each task-epoch that started 250 ms before epoch onset and ended at the epoch's conclusion (*Figure 4— figure supplement 2A*). We then identified the subset of trials (for each epoch) with zero microsaccades in the counting window (*Figure 4—figure supplement 2C*) and used these trials to recompute the cue-side preference (ROC area) of each CDh neuron. After confirming that, as in the main dataset, proportions of CDh cue-side preferences did not vary as a function of task-epoch (logistic regression, smallest tStat = 1.9143, p = 0.41), and that the interaction between task-epoch and inactivation state was non-significant (tStat = 0.8376, p = 0.33), cue-side preference values (ROC areas) were pooled across epochs for visualization and analysis (*Figure 4—figure supplement 2D-I*).

To determine whether mean ROC area in continuously isolated CDh neurons varied significantly as a function of task-epoch, we used ANOVA with two factors: (1) task-epoch, and (2) inactivation state (before/during). One ANOVA was performed on same-side inactivation data (total degrees of freedom, 'd.o.f.' = 1409; error d.o.f. = 1400; task-epoch d.o.f. = 4), and another on opposite-side (total d.o.f. = 529; error d.o.f. = 520; task-epoch d.o.f. = 4). In both ANOVAs, the effect of task-epoch was non-significant; same side: F(task-epoch) = 0.9, p = 0.98; opposite-side: F(task-epoch) = 0.63; p = 0.64.

Logistic regression was used for statistical hypothesis tests on proportions of CDh neurons with significant cue-side preferences. This mode of analysis was applied (separately) to both continuously

isolated and independent subpopulations, and separately to same-side and opposite-side inactivations. Each regression model included cue-side (cue-contra or cue-ipsi), task-epoch (1-5), and inactivation state (before or during) as categorical predictors, yielding 20 observations and 12 error degrees of freedom. For both subpopulations, same-side regression models resulted in a significant improvement over a constant model (continuously isolated: $\chi^2$ vs constant model = 93.1, p = $9.51 \times 10^{-12}$; independent: $\chi^2$ vs constant model = 109, p = $1.19 \times 10^{-20}$), but opposite-side models did not (continuously isolated: $\chi^2$ vs constant model = 28.6, p = 0.073; independent: $\chi^2$ vs constant model = 9.24, p = 0.236). All models indicated that proportions of cue-side preferences did not vary as a function of task-epoch (smallest tStat = 1.4867, smallest p = 0.14), and that the interaction between task-epoch and inactivation state was non-significant (smallest tStat = 0.5275, smallest p = 0.17).

Bootstrap tests were used to compare the distributions of cue-side preference ROC area values before and during SC inactivation. We computed the mean and skewness of ROC area distributions, then computed 95% confidence intervals by resampling with replacement 10000 times from measured ROC values to build distributions of mean and skewness and then calculating the 2.5th and 97.5th percentiles of those distributions; this was done separately on same-side and opposite-side inactivation data, and separately on before and during data. A significant difference in mean or skewness (noted as p < 0.05) was determined by finding non-overlapping confidence intervals. We note the effects of same-side SC inactivation on mean and skewness of ROC distributions in the results section; opposite-side SC inactivation caused a small but significant increase in mean (before: μ = 0.494, 95% CI = [0.488, 0.499]; during: μ = 0.506, 95% CI = [0.500, 0.512]; p < 0.05), and a non-significant change in skewness from negative to positive (before: skewness = −0.12, 95% CI = [−1.08, 0.42]; during: skewness = 0.12, 95% CI = [−0.17, 0.44]; p > 0.05).

Pearson correlation values were computed from trial-by-trial, z-scored spike counts between all pairs of simultaneously recorded neurons. Sessions with same-side SC inactivation plus recording yielded $n_{before}$ = 4280 and $n_{during}$ = 4042 pairs; opposite-side sessions yielded $n_{before}$ = 2302 and $n_{during}$ = 1468 pairs. Pairwise correlations were computed in each task-epoch, separately for cue-contra and cue-ipsi conditions, and separately for before and during SC inactivation data. To quantify any variation in distributions of correlation values as a function of epoch, cue-side, inactivation state (before/during), or their interactions, we simultaneously fit distributions in each combination of conditions using the Stable family of distributions (*Mandelbrot, 1960*). This distribution has four parameters that govern its shape (α, β, c, and μ), which allows variation in the distribution's central-tendency, width, skewness and the sharpness of its peak. To fit, we identified a single maximum likelihood solution b = [$b_\alpha$ $b_\beta$ $b_c$ $b_\mu$] to the equations: α = $Xb_\alpha$, β = $Xb_\beta$, c = $Xb_c$, and μ = $Xb_\mu$, where X is a categorical predictor array of experimental conditions including a constant term, main effects (epoch, cue-side, inactivation state), 2-way interactions, and 3-way interactions. We identified this solution (b) by minimizing the cost function: C = $\Sigma$-log(p(y| α, β, c, μ)) where y is a vector of all correlation values, and p is the Stable distribution probability density function (PDF). We computed b separately for correlations from same-side and opposite-side SC inactivations, and determined statistical significance by bootstrapping on b (we built distributions by shuffling and refitting 10000 times).

To statistically test the effect of SC inactivation on individual CDh neuron firing rates, ROC area was used. For each neuron, an average spike rate was computed in each trial before and during SC inactivation, and the distribution of average firing rates before SC inactivation (the 'noise' distribution) was compared to the distribution during inactivation (the 'signal' distribution) by computing ROC area. A neuron's firing rate during SC inactivation was considered significantly different from the rate before SC inactivation if the 95% confidence interval (CI) on ROC area did not contain (⊄) 0.5. The 95% CI was a 'permutation test' CI determined by shuffling 'signal' and 'noise' labels on firing rate values, recomputing ROC area 10000 times to build a distribution, and identifying the 2.5th and 97.5th percentiles of the distribution as the lower and upper limits of the CI, respectively.

To statistically test the effects of SC inactivation on CDh neuron firing rates in continuously isolated and independent subpopulations within and across sessions, we fit ISI distributions with a generalized linear model (GLM) using inactivation state (before/during), session ID (1-13), unit ID, subpopulation ID (continuously isolated or not), task-epoch, and their interaction terms as categorical predictors. We assumed that ISIs were gamma distributed, and accordingly used a (canonical) negative inverse link function. We fit separate GLMs to same-side and opposite-side data, and both

explained significantly more variance than a constant model (p << 0.01). The same-side GLM included 476558 observations with 476395 error degrees of freedom, and opposite-side included 200168 observations with 200103 error degrees of freedom. In the Results section, we report p-values from the main effect of inactivation state as indicating the significance of an across-sessions effect, p-values from the interaction of inactivation state and individual session ID predictors as indicating the significance of a within-session effect, and p-values from the interaction of inactivation state and subpopulation ID as indicating the significance of an effect specific to the continuously isolated subpopulation. In addition to effects reported in the Results section, we also found no significant interaction between inactivation state and task-epoch in either same-side or opposite-side GLMs (smallest tStat = 0.7914; smallest p = 0.26).

To determine whether SC inactivation affected CDh population-level encoding of cue-side and task-epoch, we used a multi-dimensional classifier approach. We treated CDh neuronal data before and during SC inactivation separately, resulting in 26 datasets (before/during x 13 sessions), and always computed 5-fold cross-validated classifier performance (*Hastie et al., 2013*). We examined the performance of several types of classifier across datasets: (1) Linear discriminant, (2) Support Vector Machine (SVM), (3) naïve bayes (NB), (4) decision tree (DT), (5) k nearest neighbor (KNN), (6) boosted decision tree (bDT), (7) boosted KNN (bKNN); each time a classifier was trained and tested on a dataset we used a Bayesian hyperparameter optimization procedure (*Snoek et al., 2012*). We selected the boosted decision tree classifier, using the AdaBoost method (*Freund and Schapire, 1997*), because we found it had the best performance across datasets. Specifically, in 13/13 'before inactivation' datasets, the bDT classifier yielded the highest correct-classification rate. In 11/13 'during inactivation' datasets the bDT had the highest correct-classification rate; in the remaining 2/13 'during inactivation datasets', the NB classifier had the highest correct-classification rate. However, in both of these 2/13 cases, the NB correct- classification rate 'during inactivation' was significantly smaller than the 'before inactivation' bDT correct- classification rate, indicating that regardless of the classifier chosen, the information available for classification during same-side SC inactivation was always smaller than before same-side SC inactivation. For completeness, we here provide the relevant correct-classification rates for the 2/13 sessions described above: session 1, before inactivation: bDT = 59% (897/1515; 95% CI = [57%, 62%]), during inactivation: bDT = 35% (913/2643; 95% CI = [33%, 36%]), NB: 42% (1114/2643; 95% CI = [40%–44%]); session 2, before inactivation bDT = 58% (1167/2015; 95% CI = [56%, 60%]), during inactivation: bDT = 35% (1085/3086; 95% CI = [33%, 37%]), NB = 43% (1338/3086; 95% CI = [42%–45%]).

The classifier-based approach allows the quantification of population-level neuronal information content that cannot be predicted by or reduced to individual or pairwise neuronal measures. However, it is important to note this does not mean changes in classifier performance (as observed during same-side inactivations) can be uniquely attributed to population-specific factors such as covariance structure and the geometrical relationship between covariance and mean. Instead, changes in classifier performance reflect changes in population-level information content that may be attributable to a mix of population-specific factors and factors visible at the individual neuron level (i.e., mean response).

## Acknowledgements

We thank B Averbeck, B Cumming, S Goldstein, L Katz, K McAlonan, C Quaia, L Wang, and G Yu for helpful discussions. This work was supported by the National Eye Institute Intramural Research Program at the National Institutes of Health.

## Additional information

### Funding

| Funder | Grant reference number | Author |
| --- | --- | --- |
| National Eye Institute | 1ZIAEY000511 | Richard J Krauzlis |
| European Research Council SYNERGY Grant scheme | 610110 | Fabrice Arcizet |

| Agence Nationale de la Recherche | RHU LIGHT4DEAF ANR-15-RHU-0001 | Fabrice Arcizet |
| Agence Nationale de la Recherche | LABEX LIFESENSES ANR-10-LABX-65 | Fabrice Arcizet |
| Agence Nationale de la Recherche | IHU FOReSIGHT ANR-15-RHU-0001 | Fabrice Arcizet |

The funders had no role in study design, data collection and interpretation, or the decision to submit the work for publication.

### Author contributions

James P Herman, Conceptualization, Data curation, Software, Formal analysis, Visualization, Writing - original draft, Writing - review and editing; Fabrice Arcizet, Conceptualization, Writing - review and editing; Richard J Krauzlis, Conceptualization, Resources, Supervision, Funding acquisition, Methodology, Writing - review and editing

### Author ORCIDs

James P Herman (ID) https://orcid.org/0000-0001-6916-2807

### Ethics

Animal experimentation: All experimental protocols (#NEI-649) were approved by the National Eye Institute Animal Care and Use Committee, and all procedures were performed in accordance with the United States Public Health Service policy on the humane care and use of laboratory animals.

### Decision letter and Author response

Decision letter https://doi.org/10.7554/eLife.53998.sa1
Author response https://doi.org/10.7554/eLife.53998.sa2

## Additional files

### Supplementary files

• Transparent reporting form

### Data availability

Data for the main figures are available via Dryad (https://doi.org/10.5061/dryad.xd2547dcx).

The following dataset was generated:

| Author(s) | Year | Dataset title | Dataset URL | Database and Identifier |
| --- | --- | --- | --- | --- |
| Herman JP, Arcizet F, Krauzlis RJ | 2019 | Data from: Attention-related modulation of caudate neurons depends on superior colliculus activity | https://doi.org/10.5061/dryad.xd2547dcx | Dryad Digital Repository, 10.5061/dryad.xd2547dcx |

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
