## [Decision Letter]

**Acceptance summary:**

The authors have addressed the reviewers' previous concerns by additional analysis and revising the manuscript. Importantly, the authors now present evidence that the results hold even when trials with and without microsaccades are separately analyzed. This study showed that inactivation of superior colliculus (SC) caused an impairment of attention-related modulations of neuronal activity in the caudate nucleus head (CDh) in a spatial attention task. Overall, the results provide great insights into the mechanism underlying attentional modulation of CDh neurons, as well as the SC's role in regulating striatal activity.

**Decision letter after peer review:**

Thank you for submitting your article "Attention-related modulation of caudate neurons depends on superior colliculus activity" for consideration by *eLife*. Your article has been reviewed by three peer reviewers, and the evaluation has been overseen by a Reviewing Editor and Michael Frank as the Senior Editor. The following individual involved in review of your submission has agreed to reveal their identity: Masayuki Matsumoto (Reviewer #2).

The reviewers have discussed the reviews with one another and the Reviewing Editor has drafted this decision to help you prepare a revised submission.

Summary:

This study examined the role of superior colliculus (SC) in attention-related modulations of neuronal activity in the caudate nucleus head (CDh) in rhesus macaques. The authors inactivated the SC while recording from the CDh in a spatial attention task. The authors found that SC inactivation altered attention-related modulation of CDh activities. This effect was evident selectively when SC inactivation was on the same side of CDh neurons. A classification analysis showed that information about the cue location and epoch identity was reduced by SC inactivation on the same side.

All the reviewers thought that this study is timely, given the increasing interest in the role of subcortical structures in attention, and provides very interesting data. However, the reviewers raised some technical and interpretational concerns that need to be addressed before publication of this work in *eLife*.

Essential revisions:

1) It is more informative if the authors can directly compare the activity of the same neuron before and during inactivation. However, from the presented data, it is unclear whether the authors were able to hold recording of single neurons over these periods. It is important that the authors present some quantitative analysis of the isolation and stability of spike recording (e.g. based on spike waveforms, amplitudes, refractory period violation etc.). Furthermore, it will be very informative if the authors present the results of the main analysis using a subset of neurons for which they are confident about the isolation and stability of recording based on some metric.

2) SC inactivation altered the animals' behavioral performance. As reviewer 1 pointed out, it remains unclear whether the authors can exclude the possibility that behavioral alterations indirectly affected the CDh neuronal activities. If the authors cannot completely eliminate this possibility, this issue should be discussed as a potential caveat of the experiment clearly in Discussion.

The reviewers raised other important issues. These additional issues as well as more detailed descriptions and suggestions for the above concerns can be found in their individual comments appended below. We hope that you can address all of these concerns, in which case we are happy to re-review a revised manuscript.

Reviewer #1:

The study by Herman and colleagues on "Attention-related modulation of caudate neurons depends on superior colliculus activity" reports several alterations in modulatory effects of neurons in the caudate nucleus related to attention after inactivation of the superior colliculus. Specifically, these alterations were (i) reductions in attentional modulation of spike rates, and (ii) the classification of task-related epochs. The results were specific to same-side (i.e. same-space) inactivation. The results are interpreted and framed in terms of a direct functional relationship between the two structures.

These are interesting results on a structure that, while long implicated in other cognitive operations, has only recently been implicated in selective attention. My main concerns relate to the framing of the study in terms of causal impacts of SC inactivation on caudate neurons given that the animals' behavior was significantly compromised by the inactivation, as elaborated below.

1) Alterations of behavior: The SC inactivation led to significant reductions in hit rates and increases in false response rates in the visual space affected by the inactivation in caudate neurons. It seems to be a logical consequence that a structure that is typically modulated by attention will reflect reductions in attentive behavior to certain parts of visual space.

2) Framing of the study: Given the behavioral effects, in principle, the alterations in neuronal activity and differences in attentional modulation effects may be a direct consequence of the altered behavior rather than of any specific functional relationship between the two structures. In order to interpret the data in terms of functional relationships between SC and basal ganglia, this relationship should be established first in simultaneous recordings from the two structures that would shed light on their interactions in a more mechanistic way. Once these interactions are understood, it might be more straightforward to interpret the inactivation effects in terms of reflecting specific functional relationships. Alternatively, the study will need different framing.

3) Specificity of inactivation effects: It is laudable to demonstrate that the inactivation effects pertain to specific parts of visual space. However, it is not clear whether they are specific to the inactivated structure. Any network alteration that alters behavior may lead to similar effects in caudate neural response patterns (e.g. FEF inactivation?). Please discuss.

4) Figure 2: Please show population data in the same format as the example unit.

5) Please show all group/population data separately for the two animals in supplementary figures to evaluate the consistency of the effects.

6) How many trials were recorded during SC inactivation? How did the animal's behavior change over the course of individual sessions? Often animal behavior is less good during the second half of a recording session. Could that have negative impacts on the results and confound any?

Reviewer #2:

The caudate nucleus has recently attracted attention as a part of the neural network underlying the control of visual attention. The present study attempted to identify the source of the attention signal in the caudate nucleus. A candidate is the superior colliculus (SC) that sends indirect inputs to the head of the caudate nucleus (CDh) and is also known to play a crucial role in controlling visual attention.

The authors recorded the activity of CDh neurons in monkeys performing a visual attention task before and during SC inactivation. They found that the neuronal modulation of the CDh evoked by visual attention was reduced during SC inactivation compared with the modulation before the inactivation. The authors conducted other analyses as well and observed that, for example, decoding task states from CDh neuronal activity went well before SC inactivation but was impaired during the inactivation. Based on these findings, the authors concluded that the SC is a major candidate of the source of the attention signal in the CDh.

The theme, i.e., the circuit-level mechanism underlying visual attention, is attractive. The experiment was well designed. The story and conclusion are logical. But, I have a couple of concerns about analyses. Before decision, I would like to see their responses to the following my comments.

1) A key physiological feature of SC neurons, which seems vital for spatial attention, is their topographically organized response field. Orienting attention to a specific visual location would require SC neurons with a response field corresponding to that location. However, the authors did not analyze the relationship between the effect of SC inactivation on the attentional-related CDh neuronal modulation and the response field of inactivated SC neurons. As described in the manuscript, the authors examined the effect of SC inactivation on visually-guided saccades. In particular, they examined the spatial extent of saccade deficits, from which the response field of inactivated SC neurons can be estimated. I expect that, if the response field of inactivated SC neurons was close to the spatial cue location, the effect of the SC inactivation on the attention-related CDh neuronal modulation was larger. On the other hand, if the response field of inactivated SC neurons was far from the spatial cue location, the effect was smaller. I think such analysis makes their conclusion stronger.

2) Inactivating unilateral SC may cause fixation problems. For example, the SC inactivation may cause involuntary or micro saccades to the ipsilateral visual field. Such ipsilateral saccades could reduce the effect of the attention-related CDh neuronal modulation to the contralateral visual field. The authors need to confirm that such saccadic effects were not evoked by the SC inactivation.

3) The simple and effective analysis to demonstrate the effect of SC inactivation on the CDh activity is the comparison of attention-related CDh neuronal modulation between "before" and "during" SC inactivation. The authors need to show how many CDh neurons exhibited a significant decrease in the attention-related neuronal modulation during SC inactivation comparted with before the inactivation. This is the simplest analysis. I did not find such single-neuron-level analysis throughout the manuscript. They just showed single-neuron examples.

4) Even at the population level analysis to compare the attention-related CDh neuronal modulation between "before" and "during" SC inactivation, the authors used an unusual comparison procedure. In Figure 3E and F, they exhibited the distributions of the attention-related CDh neuronal modulations in each epoch of "before" and "during" SC inactivation. They just analyzed the shift of the distributions. However, they need to use "paired" comparison procedure for each neuron.

Reviewer #3:

Herman et al. examined the effects of unilateral SC inactivation on caudate activity during a covert attention task. They found that SC inactivation altered the caudate neuron's responses for attentional cue location and reduced the interneuronal correlation. These effects depended on whether the inactivated SC was on the same or opposite side of caudate neurons. They also performed a classification analysis to show that classification performance for the cue location and epoch identity was impaired with SC inactivation for the same-side condition.

The authors performed a very challenging experiment aptly. The presence of a causal effect of SC on attentional modulation of caudate activity is convincing. I have some concerns about the details and interpretation of specific effects that the authors reported, which warrant additional analyses and/or text revision.

– The text in the subsection “Effects of SC inactivation on CDh spatial cue preferences” has some abrupt logical jumps.

1) Figure 2 purported to show that the effects of SC inactivation on CDh activity depended on whether the same-side SC was inactivated. This would be best illustrated by showing the same example neuron with same-side and different-side SC inactivation. The authors presented two CDh neurons, for the two kinds of SC inactivation, respectively. It is inconclusive whether the different effects were due to which SC was inactivated or which CDh neuron was recorded.

2) The authors stated that "inactivation of the opposite-side SC produced small decreases in the number of CDh neurons with ipsi-cue preference" and summarized the paragraph by stating that "…these reductions were largest for CDh neurons with contra-cue preferences recorded on the same side as the inactivation." Figure 2 does not support these statements.

– I appreciate the authors' caution on asserting the "same" neuron was recorded before and during inactivation. But if there were recordings with stable waveforms throughout a session, data from these neurons should be presented. For example, if muscimol injection in SC caused a consistent shift in the relative positions of neural tissue and electrode contacts, the interpretation of inactivation effects needs to include the possibility that different sub-populations were recorded before and after inactivation. The difference in SC inactivation for "same-side" and "opposite-side" might be due to the latter causing less physical movements of the brain tissue relative to recording contacts.

– Although the differences between "Same-side SC" and "Opposite-side SC" in Figure 3 are clear, I am still concerned whether these differences were due to which side of SC was inactivated or which subpopulation of CDh neurons were recorded. While Figure 3A showed cue-contra dominance, Figure 3C showed cue-ipsi dominance before inactivation. Where in the caudate were the sets of recording tracks for Figure 3A and C?

– The authors presented summary statistics for the effects on SC inactivation on CDh firing rates (subsection “Changes in spike-count correlations of CDh neurons”). Such data are very important to assess the relationship between SC and CDh activity and should be presented in more details. For example, is there a difference in firing rates for a specific epoch? Figure 2A and B showed a clear increase in the neuron's response to motion patches and during the patch-change epoch. How prevalent are similar changes in the population? How prevalent are similar changes in the subpopulation of neurons with changes in cue-preference? Was the apparent lack of effect on firing rates due to truly no effect or inconsistent effects for different neurons? What model could explain a removal of common inputs to CDh neurons by SC inactivation and lack of any effect on these neurons' firing rates?

– I am confused by the classification results. As I understand it, the authors showed that SC inactivation changed the relative proportion of cue-ipsi/cue-contra preferences of CDh neurons, but not the fraction or modulation strength (ROC area) of neurons showing some preference (Figure 3). Why would there be an increase in classification errors for cue-side with SC inactivation? The authors also stated that the SC inactivation-induced changes did not differ among epochs. Why would the cue-side classification errors differ so much between epochs 1/2 and the other epochs? Judging from Figure 3F and H, the fraction and ROC area of cue-side-preferring neurons were similar during SC inactivation for same-side and opposite-side. Why was the classification performance so much worse for same-side than for opposite-side SC inactivation?

– The authors chose the classifier based on the overall best performance for all datasets. How would the classification results change, if the authors used the best classifier for each dataset? This approach is better to assess the maximal information present in each dataset, without having to account for task-irrelevant differences in the datasets that could affect classification performance.

– At a higher level, I don't get the rationale for testing how covert spatial attention changes the involvement of CDh in encoding non-spatial epoch information. A diagram/model showing the authors' assumptions would be helpful.

[Editors' note: further revisions were suggested prior to acceptance, as described below.]

Thank you for resubmitting your article "Attention-related modulation of caudate neurons depends on superior colliculus activity" for consideration by *eLife*. Your revised article has been reviewed by three peer reviewers, and the evaluation has been overseen by a Reviewing Editor and Michael Frank as the Senior Editor. The following individual involved in review of your submission has agreed to reveal their identity: Masayuki Matsumoto (Reviewer #2).

The reviewers have discussed the reviews with one another and the Reviewing Editor has drafted this decision to help you prepare a revised submission.

As you will see below, the reviewers thought that the authors have addressed most of the concerns adequately. However, there remain some concerns regarding the strength of the main conclusions. Specifically, although the authors now make some arguments to support the point that the observed effects are relatively direct effects of inactivation of the superior colliculus. However, we thought that it is difficult to conclude it with the presented data. Given the importance of the experiment and the data, we would like to proceed toward publication of this work. However, we would like the authors to weaken these conclusions and make more balanced discussions. Furthermore, reviewer 1 thought that the authors should present quantitative analysis on eye movements, given that the authors must have these data (please see below). We would like to see the authors' response to these points before making a final decision.

Reviewer #1:

This is a re-review of the study by Herman and colleagues on "Attention-related modulation of caudate neurons depends on superior colliculus activity". The authors have addressed many of the reviewers' concerns in satisfactory ways. My general main concern regarding the approach remains: First, it is difficult to interpret the effects of inactivating a structure on another structure without understanding the physiological interactions in the first place. Second, perturbing a part of a large-scale network will cause interruptions other than in the caudate nucleus, and these can influence or even account for what has been studied here. So, in the end, I am not clear what many of the results mean in a broader context. Still, given the scarcity of studies on the caudate and attention, I am supportive of publication, but I find the interpretations still overreaching including the title – perhaps a bit more caution might be helpful.

Another point that remains and that needs to be addressed are the eye movement concerns raised by reviewer #2. Since the AUs have the data, it would be preferable to address the concern by showing compelling data rather than arguing that the influence of microsaccades is unlikely to play a role here. Microsaccades have been shown to play major roles in influencing attentional modulation, and I feel this issue can be addressed in better ways.

Reviewer #2:

The authors adequately responded to all of my concerns. Although Figure 3B was lost in the main manuscript, I found it near the end of the manuscript as an individual figure. I recommend its publication in *eLife*.

Reviewer #3:

The additional analyses and text revisions addressed my concerns.

---

## [Author Response]

Essential revisions:1) It is more informative if the authors can directly compare the activity of the same neuron before and during inactivation. However, from the presented data, it is unclear whether the authors were able to hold recording of single neurons over these periods. It is important that the authors present some quantitative analysis of the isolation and stability of spike recording (e.g. based on spike waveforms, amplitudes, refractory period violation etc.). Furthermore, it will be very informative if the authors present the results of the main analysis using a subset of neurons for which they are confident about the isolation and stability of recording based on some metric.

We agree that direct comparison of the activity of the same neurons before and during SC inactivation would be informative. To this end, based on the assumption that individual neurons can be reliably identified by quantitative metrics of spiking activity, and building on established methods for such identification, we identified a

“continuously isolated” subset of recorded CDh neurons that were stably isolated across both “before” and “during” phases of each inactivation-plus-recording session. We describe our analysis of “continuously isolated” CDh neurons (as well as the analysis of “independent” CDh neurons, which does not require these assumptions) in the Results section (subsection “Effects of SC inactivation on CDh cue location preferences”) and delineate the identification procedure in the Materials and methods section (subsection “Spike Sorting and Unit Selection”).

An analysis of cue-side preferences of continuously isolated CDh neurons before and during SC inactivations yielded qualitatively the same results that we found in the population of CDh neurons at large: “same-side” SC inactivation caused a reduction in the prevalence of cue-contra preferences and an increase of cue-ipsi while “opposite-side” SC inactivation caused no statistically significant change. The effects of SC inactivations on “continuously isolated” CDh neurons are now described in the Results section (subsection “Effects of SC inactivation on CDh cue location preferences”), “same-side” inactivation effects are presented in a new Figure 3 and “opposite-side” in Figure 3—figure supplement 2.

We also analyzed the effects of SC inactivation on individual “continuously isolated” CDh neuron firing rates, finding that “same-side” SC inactivations had a greater tendency to increase individual firing rates (Figure 3—figure supplement 1G), whereas “opposite-side” inactivations caused no systematic change – a comparable number of CDh neurons increased and decreased their firing rate (Figure 3—figure supplement 1H). These findings are described in the Results section (subsection “Changes in spike-count correlations of CDh neurons”) and the approach is described in Materials and methods (subsection “Data analysis”). We also corrected an error in the values reported for population average firing rates in the Results section (subsection “Changes in spike-count correlations of CDh neurons”).

2) SC inactivation altered the animals' behavioral performance. As reviewer 1 pointed out, it remains unclear whether the authors can exclude the possibility that behavioral alterations indirectly affected the CDh neuronal activities. If the authors cannot completely eliminate this possibility, this issue should be discussed as a potential caveat of the experiment clearly in Discussion.

As suggested, we have added a new section to the Discussion to discuss this potential caveat. We now clarify why we consider it unlikely that changes in CDh neuronal activity result indirectly from behavioral effects of SC inactivation in the Discussion section (subsection “Could CDh neuronal changes during SC inactivation be indirect or secondary effects?”).

In brief, we consider a direct circuit-level explanation, rather than an indirect behavioral effect, most likely because the changes in cue-side preferences of CDh neurons do not always match the behavioral effects caused by SC inactivation but instead depend on which side of the circuit is manipulated. Same-side SC inactivations reduced cue-contra hit rate (Figure 1C) and reduced CDh cue-contra preferences (new Figure 3A, B), but opposite-side SC inactivations reduced cue-ipsi hit rate (Figure 1C) without affecting CDh cue-ipsi preferences (Figure 3—figure supplement 2A, B). This shows that the changes in cue-side preferences of CDh neurons were not dictated by the behavioral impairments caused by SC inactivation. Instead, these results are consistent with the anatomical circuits that would allow SC output to influence CDh, which are strongly ipsilaterally biased (discussed in the subsection “A reversal of the classic subcortical hierarchy”).

Although this dissociation of cue-side effects between behavior and neurons may be surprising, previous work from our lab has shown that SC inactivation can impair attention-task behavior while attention-related modulation of the attended visual feature in sensory cortex remains totally intact (Zénon and Krauzlis, 2012). This finding demonstrates that altered attention-task behavior does not guarantee altered attention related modulation of neuronal activity, consistent with our current results. We now briefly comment on this point in the Discussion section (subsection “Could CDh neuronal changes during SC inactivation be indirect or secondary effects?”).

The reviewers raised other important issues. These additional issues as well as more detailed descriptions and suggestions for the above concerns can be found in their individual comments appended below. We hope that you can address all of these concerns, in which case we are happy to re-review a revised manuscript.Reviewer #1:[…] My main concerns relate to the framing of the study in terms of causal impacts of SC inactivation on caudate neurons given that the animals' behavior was significantly compromised by the inactivation, as elaborated below.1) Alterations of behavior: The SC inactivation led to significant reductions in hit rates and increases in false response rates in the visual space affected by the inactivation in caudate neurons. It seems to be a logical consequence that a structure that is typically modulated by attention will reflect reductions in attentive behavior to certain parts of visual space.

In our response to Essential revisions #2 above, we clarify why we consider it most likely that a direct functional relationship between SC and CDh is the cause of changes in CDh neuronal activity during SC inactivation. We now include a discussion of this point in a new section in the Discussion (subsection “Could CDh neuronal changes during SC inactivation be indirect or secondary effects?”). In brief, the changes we observed in CDh neuronal responses during SC inactivation are most consistent with those changes being caused by a direct functional relationship (between SC and CDh) rather than an indirect behavioral impairment.

2) Framing of the study: Given the behavioral effects, in principle, the alterations in neuronal activity and differences in attentional modulation effects may be a direct consequence of the altered behavior rather than of any specific functional relationship between the two structures. In order to interpret the data in terms of functional relationships between SC and basal ganglia, this relationship should be established first in simultaneous recordings from the two structures that would shed light on their interactions in a more mechanistic way. Once these interactions are understood, it might be more straightforward to interpret the inactivation effects in terms of reflecting specific functional relationships. Alternatively, the study will need different framing.

We agree with the reviewer that simultaneous recordings in SC and basal ganglia would be very informative, although that work is beyond the scope of our current study. We also think that our current results, showing a causal relationship between the two structures during an attention task, provides a strong rationale for conducting such simultaneous recordings. We hope the reviewer shares our appreciation for the value of causal manipulations and that our Introduction and framing of the study is appropriate.

3) Specificity of inactivation effects: It is laudable to demonstrate that the inactivation effects pertain to specific parts of visual space. However, it is not clear whether they are specific to the inactivated structure. Any network alteration that alters behavior may lead to similar effects in caudate neural response patterns (e.g. FEF inactivation?). Please discuss.

The reviewer raises an interesting point that we took for granted in the original manuscript. We agree that even our restricted, focal inactivation of the SC will cause network-wide changes in activity, and that it might be possible to achieve comparable alterations in network activity by manipulations in other structures. However, the SC is an especially useful structure to target because focal SC inactivation causes large and easily reproducible spatially specific deficits in attention-task performance. We have also done FEF inactivations (Bollimunta et al., 2018) and the effects on selective attention are smaller and less reliable. Although the reviewer recognizes that the CDh is part of these circuits shared with the SC and FEF, the possible importance of CDh circuits for selective attention is not yet generally understood. It is therefore important to establish this point directly by showing a causal relationship with attention-related circuits, and we think the SC provides the best opportunity to doing so. We now briefly discuss this point about extended network effects in the Discussion section (subsection “Could CDh neuronal changes during SC inactivation be indirect or secondary effects?”).

4) Figure 2: Please show population data in the same format as the example unit.

We have followed the reviewer’s suggestion, but with a modification. Because the response profiles and cue-side preferences of CDh neurons are heterogeneous, averaging across neurons yields curves that are quite flat. For this reason, we chose not to present population average curves in the main text. Instead, we have included a supplementary figure that includes population averages as well as “image” plots in which one row of pixels represents the activity profile of a single neuron, with pixel intensity representing activity magnitude (Figure 3—figure supplement 1I-P). The “image” plots illustrate the heterogenous response profiles across the population that are obscured by population averages.

5) Please show all group/population data separately for the two animals in supplementary figures to evaluate the consistency of the effects.

As requested, we have added figures showing data from each of the two animals separately in the same format used in the main text (Figure 3—figure supplements 3-4, Figure 4—figure supplements 1-2, Figure 5—figure supplement 1, Figure 5, Figure 6—figure supplements 1-2). We have provided these animal-specific figures for the results presented in Figures 3-6.

6) How many trials were recorded during SC inactivation? How did the animal's behavior change over the course of individual sessions? Often animal behavior is less good during the second half of a recording session. Could that have negative impacts on the results and confound any?

During inactivation, we collected a minimum of 544 trials and a maximum of 952 trials

(subsection “Neuronal Recordings and Inactivation”). To quantify the stability of each animal’s behavior in each session (independent of SC inactivation effects), we subdivided the “before” and “during” portions of each session into two halves and computed hit rate (separately for each cue-side condition) in each half. This yielded 8 hit rates for each session ([first half of before, second half of before, first half of during, second half of during] X 2 cue-side conditions); we then compared hits in the first half to the second half separately for both cue-side condition and for before / during using χ^2^-proportion tests (4 tests per session). Of the 26 “before” portions examined this way (13 sessions X 2 cue-side conditions), we identified 6 with significantly different hit rates in the first half compared to the second half (all p < 0.05; largest χ^2^ = 15.6564); of the 26 “during” portions, 3 showed significantly different rates (all p < 0.05; largest χ^2^ = 6.2013). These 9 cases were roughly equally likely to occur in same side inactivations (6 / 36 portions; 17%) and opposite side inactivations (3 / 16 portions; 19%). However, in all 9 cases, hit rate was *larger* in the second half compared to the first half (of either before *or* during portion). Also, importantly, the 3 cases in which performance improved significantly during SC inactivation were conditions in which the spatial cue was presented outside the inactivation-deficit area. This suggests that any effects observed during SC inactivation are unlikely to be attributable to a lack of motivation or engagement with the task on the part of the animal. In addition, it is unlikely that behavioral drift can explain or confound our results because same-side and opposite-side SC inactivation sessions proceeded on the same time-course and yielded totally different patterns of CDh neuronal data during SC inactivation compared to before inactivation. We now briefly comment on these behavioral stability metrics in the Materials and methods section (see the aforementioned subsection).

Reviewer #2:[…] The theme, i.e., the circuit-level mechanism underlying visual attention, is attractive. The experiment was well designed. The story and conclusion are logical. But, I have a couple of concerns about analyses. Before decision, I would like to see their responses to the following my comments.1) A key physiological feature of SC neurons, which seems vital for spatial attention, is their topographically organized response field. Orienting attention to a specific visual location would require SC neurons with a response field corresponding to that location. However, the authors did not analyze the relationship between the effect of SC inactivation on the attentional-related CDh neuronal modulation and the response field of inactivated SC neurons. As described in the manuscript, the authors examined the effect of SC inactivation on visually-guided saccades. In particular, they examined the spatial extent of saccade deficits, from which the response field of inactivated SC neurons can be estimated. I expect that, if the response field of inactivated SC neurons was close to the spatial cue location, the effect of the SC inactivation on the attention-related CDh neuronal modulation was larger. On the other hand, if the response field of inactivated SC neurons was far from the spatial cue location, the effect was smaller. I think such analysis makes their conclusion stronger.

We agree that it is important to consider how variability in the extent of the SC inactivation might have been related to variation in the effects on CDh neurons. To address this point, we now include each session’s inactivation-deficit map in a supplementary figure (Figure 1—figure supplement 2), documenting the session-by-session overlap of SC inactivation-deficit area with the location of the spatial cue. The method used to generate these maps is described in the Materials and methods section (subsection “Data analysis”). Consistent with inactivation-deficit maps estimated online in each session, we found that the visual field area affected by SC inactivation always completely covered the area occupied by the spatial cue so the cued visual stimulus (in one hemifield). We now note this in the Results section (subsection “Effects of SC inactivation on attention-task performance”). Consequently, there is little meaningful variation in the relative locations of the deficit area and the spatial cue against which to compare the magnitude of inactivation effects in CDh neurons. This was our intention, since it simplifies the pooling of results across CDh neurons.

2) Inactivating unilateral SC may cause fixation problems. For example, the SC inactivation may cause involuntary or micro saccades to the ipsilateral visual field. Such ipsilateral saccades could reduce the effect of the attention-related CDh neuronal modulation to the contralateral visual field. The authors need to confirm that such saccadic effects were not evoked by the SC inactivation.

We agree that it is important to exclude “low level” effects on visual stability as possible explanations of observed effects in CDh neurons. However, we consider it unlikely that changes in microsaccade frequency or direction can explain the effects we have observed for two reasons: (1) inactivation of same-side and opposite-side SC should exert opposing effects on microsaccades, yet same-side SC inactivations had a clear effect on CDh neuronal activity but opposite-side SC inactivations did not; (2) great care was taken to ensure that muscimol injections were made away from the rostral pole of the colliculus to minimize the likelihood that inactivation affected fixation. We now explicitly discuss these points in a new paragraph in the Discussion section (subsection “Could CDh neuronal changes during SC inactivation be indirect or secondary effects?”).

3) The simple and effective analysis to demonstrate the effect of SC inactivation on the CDh activity is the comparison of attention-related CDh neuronal modulation between "before" and "during" SC inactivation. The authors need to show how many CDh neurons exhibited a significant decrease in the attention-related neuronal modulation during SC inactivation comparted with before the inactivation. This is the simplest analysis. I did not find such single-neuron-level analysis throughout the manuscript. They just showed single-neuron examples.

We agree that the direct comparison of attention-related modulation in single CDh neurons before vs. during SC inactivation is an important analysis. Please see our response to “Essential revisions #1”.

We thank Dr. Matsumoto for the suggestion. We did not include such a single-neuron level analysis in our original submission because it requires making additional assumptions about the identity of neurons before and during SC inactivation. However, prompted by the reviewer’s comment, we have now implemented what we think is a rigorous standard for identifying neurons that were most likely to be “continuously isolated” during the experiment. We have added figures (Figure 3; Figure 3—figure supplement 2) and several analyses specifically comparing attention-related neuronal modulation before and during SC inactivation in a subpopulation of CDh neurons that were “continuously isolated” across “before” and “during” inactivation experimental phases. We explain how CDh neurons were parsed into “continuously isolated” and “independent” subpopulations (subsection “Effects of SC inactivation on CDh cue location preferences”), and describe attention-related modulation in continuously isolated CDh neurons in the Results section (subsection “Effects of SC inactivation on CDh cue location preferences”).

4) Even at the population level analysis to compare the attention-related CDh neuronal modulation between "before" and "during" SC inactivation, the authors used an unusual comparison procedure. In Figure 3E and F, they exhibited the distributions of the attention-related CDh neuronal modulations in each epoch of "before" and "during" SC inactivation. They just analyzed the shift of the distributions. However, they need to use "paired" comparison procedure for each neuron.

Again, this reflected the more conservative approach we initially took about not assuming that neurons were continuously isolated. In the newly added figures (Figure 3; Figure 3—figure supplement 2), we directly compare attention-related CDh neuronal modulation before and during SC inactivation on a single-neuron basis. We also retain our original analysis, which considers the “before” and “during” populations of CDh neurons to be independent. The fact that both analyses provide the same answer gives us stronger confidence in our conclusions.

Reviewer #3:[…] The authors performed a very challenging experiment aptly. The presence of a causal effect of SC on attentional modulation of caudate activity is convincing. I have some concerns about the details and interpretation of specific effects that the authors reported, which warrant additional analyses and/or text revision.– The text in the subsection “Effects of SC inactivation on CDh spatial cue preferences” has some abrupt logical jumps.

We have attempted to improve the flow in this section of the text (subsection “Effects of SC inactivation on CDh cue location preferences”). In part, we think the problem arose because we did not explicitly explain that we were treating the ‘before’ and ‘during’ populations of neurons as independent. In response to other comments, we now include two sets of analyses – the original independent approach, and a new neuron-by-neuron analysis. We hope the new presentation is clearer.

1) Figure 2 purported to show that the effects of SC inactivation on CDh activity depended on whether the same-side SC was inactivated. This would be best illustrated by showing the same example neuron with same-side and different-side SC inactivation. The authors presented two CDh neurons, for the two kinds of SC inactivation, respectively. It is inconclusive whether the different effects were due to which SC was inactivated or which CDh neuron was recorded.

We agree that comparing the effects of same-side SC inactivation and opposite-side SC inactivation on a single, unchanging set of CDh neurons would be a powerful and dramatic illustration, but the methods available to us currently do not allow such a comparison. Specifically, the effects of muscimol persist for too long (8+ hours) to make this comparison in a single session. We are working toward using optogenetic methods to manipulate SC neuronal activity so that individual trials with one or both side of SC perturbed could be interleaved, but these methods are not yet reliable in the monkey.

Until such methods are available, we believe that our approach of suppressing activity for part of a session and including appropriate control conditions and sessions (opposite-side SC inactivations) provide a solid basis for our conclusions about attention-related spatial selectivity in CDh neurons. However, we do now include results showing data from the same neurons before and during SC inactivation, which we think strengthens the conclusions.

2) The authors stated that "inactivation of the opposite-side SC produced small decreases in the number of CDh neurons with ipsi-cue preference" and summarized the paragraph by stating that "…these reductions were largest for CDh neurons with contra-cue preferences recorded on the same side as the inactivation." Figure 2 does not support these statements.

We have clarified this section to emphasize that this sentence alludes to the yet-to-be shown population data that do support those statements (subsection “Effects of SC inactivation on CDh cue location preferences”).

– I appreciate the authors' caution on asserting the "same" neuron was recorded before and during inactivation. But if there were recordings with stable waveforms throughout a session, data from these neurons should be presented. For example, if muscimol injection in SC caused a consistent shift in the relative positions of neural tissue and electrode contacts, the interpretation of inactivation effects needs to include the possibility that different sub-populations were recorded before and after inactivation. The difference in SC inactivation for "same-side" and "opposite-side" might be due to the latter causing less physical movements of the brain tissue relative to recording contacts.

Please see our response to “Essential revisions #1”.

We agree, and a similar point was made by reviewer #2. We have added figures (figure 3; Figure 3—figure supplement 2) and analyses specifically comparing attention-related neuronal modulation before and during SC inactivation in a subpopulation of CDh neurons that were “continuously isolated” across “before” and “during” inactivation experimental phases.

We explain how CDh neurons were parsed into “continuously isolated” and

“independent” subpopulations and describe attention-related modulation in continuously isolated CDh neurons in the Results section (subsection “Effects of SC inactivation on CDh cue location preferences”).

– Although the differences between "Same-side SC" and "Opposite-side SC" in Figure 3 are clear, I am still concerned whether these differences were due to which side of SC was inactivated or which subpopulation of CDh neurons were recorded. While Figure 3A showed cue-contra dominance, Figure 3C showed cue-ipsi dominance before inactivation. Where in the caudate were the sets of recording tracks for Figure 3A and C?

We took care to ensure that recording tracks used in same-side and opposite-side SC inactivations were identical or minimally different. With the exception of 1 recording track in 1 opposite-side SC inactivation in monkey P, all recording tracks were identical across same-side and opposite-side inactivations. In the case of this one exception, the recording track was 1 mm posterior to the track used in all other sessions (in this monkey).

In monkey P, we used 3 recording tracks across experiments (referred to as tracks 1, 2, and 3). Track 1 was located at AC+5 (5 mm anterior to the anterior commissure: “AC”), track 2 at AC+6, and track 3 at AC+8. In all 5 same-side inactivations, we used tracks 2 and 3; in 1 opposite-side inactivation we used tracks 2 and 3 and in the other we used track 1 and 3. In monkey R, we used 2 recording tracks across experiments (tracks 1 and 2). Track 1 was located at AC+2 and track 2 at AC+4. In all 6 experiments in this monkey (4 same-side, 2 opposite-side inactivations), we used track 1 and track 2. We have included this information in the Materials and methods section (subsection “Neuronal Recordings and Inactivation”), and we have added a supplementary figure showing the estimated positions of recording contacts for each recording track overlaid on structural MRI images (Figure 1—figure supplement 3). Because of the minimal difference in recording tracks between same-side and opposite-side inactivations, we attribute the difference in bias in before-inactivation cue-side preferences to inherent sampling variability.

– The authors presented summary statistics for the effects on SC inactivation on CDh firing rates (subsection “Changes in spike-count correlations of CDh neurons”). Such data are very important to assess the relationship between SC and CDh activity and should be presented in more details. For example, is there a difference in firing rates for a specific epoch?

We agree that these data are important for understanding the nature of the functional relationship between SC and CDh neuronal activity. To address this, we have added several figure panels addressing the effects of SC inactivation on CDh neuron firing rates (Figure 3—figure supplement 1). To specifically address the question of whether there was an effect of SC inactivation on firing rates for a specific epoch, we revisited the GLMs used to compare firing rates before and during inactivation, and added a task-epoch predictor to the existing predictors (inactivation state, session ID, unit ID, and subpopulation ID). We found that the effect of SC inactivation on firing rate did not vary significantly with task-epoch: in both same-side and opposite-side GLMs, the interaction term “inactivation state:task-epoch” was not significant, which we report in the Materials and methods section (subsection “Data analysis”).

Figure 2A and B showed a clear increase in the neuron's response to motion patches and during the patch-change epoch. How prevalent are similar changes in the population?

Regarding responses to the onset of the motion patches (the “visual” task-epoch), 6% (9/141) of continuously isolated CDh neurons (from same side inactivations) showed a significant increase in activity as compared to the “post-cue” task-epoch (which immediately precedes that visual epoch); a significant increase in firing was identified by computing ROC area and 95% confidence interval (CI) comparing spike-count distributions in the “post-cue” epoch (the “noise” distribution) to those in the “visual” task-epoch (the “signal” distribution), and determining whether the 95% CI was greater than 0.5. However, with regard to the reviewer’s observation of an increase during the “patch-change” epoch: there was no “patch-change” epoch. As indicated in the Materials and methods section (subsection “Data analysis”), we truncated neuronal data immediately prior to the time of the motion change. We did this because we previously found that many caudate MSNs display activity related to joystick-release or stimulus-change-related activity modulated by joystick-release (Arcizet and Krauzlis, 2018). Because SC inactivation systematically altered the probability of reporting the motion-direction change, alterations in CDh activity after the stimulus change could be secondary effects caused by changes in motor behavior, so neuronal data from these later task-epochs were excluded from analysis.

How prevalent are similar changes in the subpopulation of neurons with changes in cue-preference?

Under the assumption that a “change in cue-preference” refers to a CDh neuron that had a one preference before SC inactivation (either cue-contra, cue-ipsi, or no preference) and did not have the same preference during SC inactivation, and that “similar changes” refers to a neuron displaying a significant increase in firing rate in the “visual” task-epoch as defined above, 44% (4 / 9) of continuously isolated CDh neurons that changed cue-preferences showed similar changes. Across the population of continuously isolated CDh neurons (not limited to those with a significant increase in firing rate in the visual epoch), 58% (82/141) of neurons showed a change in cue preference. This suggests that CDh neurons exhibiting phasic activation for motion-dot onset are no more likely to change cue-preference during same side SC inactivation than CDh neurons at large.

Was the apparent lack of effect on firing rates due to truly no effect or inconsistent effects for different neurons?

Good question – the effects were different across neurons, resulting in no net effect at the population level. We now detail the effects of SC inactivation on CDh firing rates at the single-neuron level (in the “continuously isolated” subpopulation), and at the population level (in both subpopulations) in the Results section (subsection “Changes in spike-count correlations of CDh neurons”). Approximately 95% (184/194) of individual continuously isolated CDh neurons showed significantly different firing rates during SC inactivation compared to before, with 103 showing an increased rate and 81 a decreased rate.

What model could explain a removal of common inputs to CDh neurons by SC inactivation and lack of any effect on these neurons' firing rates?

Prompted by the reviewer’s previous question, we now clarify that there was no net change at the population level, but that individual neurons tended to increase or decrease their firing rates during SC inactivation. This simplifies the interpretation and is also consistent with the reduction in both positive and negative spike-count correlations.

– I am confused by the classification results. As I understand it, the authors showed that SC inactivation changed the relative proportion of cue-ipsi/cue-contra preferences of CDh neurons, but not the fraction or modulation strength (ROC area) of neurons showing some preference (Figure 3). Why would there be an increase in classification errors for cue-side with SC inactivation?

The performance of multi-dimensional (multi-neuron) classifiers cannot be predicted on the basis of individual dimensions (neurons) unless the dimensions are totally independent. In neuronal data, the presence of correlated variability among neurons (dimensions) means that it is impossible to predict the performance of a multidimensional classifier on the basis of individual neuron responses. In this case, the structure of the covariation among neurons (the covariance) and its geometrical relationship to the mean responses in the conditions being classified are what determine the performance of the classifier. Unfortunately, it is impractical to directly visualize the covariance because it is highly specific to the neurons in which it is measured – unless one is recording from the exact same set of neurons in each session, it is meaningless to compare two covariances obtained in different sessions. This limitation is one of the reasons we chose to use a multi-dimensional classifier to examine the information contained in the CDh population response: these classifiers allow the quantification of differences in multi-dimensional responses that arise from *both* differences in covariance and mean response. All that said, it is worth noting that individual neuron cue-side preferences are moderately weaker during SC inactivation compared to before (Figure 4). More succinctly, the increase in classification errors for cue-side with SC inactivation can only be due to a combination of effects on “mean response”, on “covariance”, and on their geometric relationship. We have tried to concisely clarify these points in the Results section (subsection “Changes in classifier performance based on CDh neuronal activity”) and the Materials and methods section (subsection “Data analysis”).

The authors also stated that the SC inactivation-induced changes did not differ among epochs. Why would the cue-side classification errors differ so much between epochs 1/2 and the other epochs?

Because cue-side presentation is blocked, and because there is little variability in the time of cue presentation relative to initial fixation, the animal has a strong expectation that spatial cue is on the verge of being presented immediately after initial fixation (in the pre-cue task-epoch). Meanwhile, the post-cue epoch includes a portion of the time that the spatial cue is visible. This means that there are two strong signals related to cue presentation in these two epochs that are not present (or are present more weakly) in the other epochs. Note also that it is not cue-side classification performance alone that is superior in these task-epochs, but classifier performance overall. This means that the classifier is better able to uniquely identify these epochs compared to others. It may be that because the visual stimulus information present in the “visual”, “delay”, and “prechange” task-epochs is extremely similar, that the population-level differences that would allow the classifier to discriminate amongst them is reduced; this assertion is supported by the observation that the classifier almost never misclassifies the pre-cue or post-cue task-epochs as being the “visual”, “delay”, or “pre-cue” task-epoch (Figure 6C, D, I, J).

Judging from Figure 3F and H, the fraction and ROC area of cue-side-preferring neurons were similar during SC inactivation for same-side and opposite-side. Why was the classification performance so much worse for same-side than for opposite-side SC inactivation?

As discussed above, the classifier performance is influenced by the “mean response”, “covariance”, and the geometric relationship between mean and covariance. Changes in mean response would manifest as a change in ROC area, but changes in covariance and mean-covariance-geometry not necessarily lead to changes in ROC area, but could degrade classifier performance. We found that the “mean response” (Figure 4I) and spike-count correlations (Figure 5; strongly suggestive of covariance changes) were affected by same-side SC inactivation, but not by opposite-side SC inactivation (Figure 4J), consistent with the difference in classifier performance between these conditions.

– The authors chose the classifier based on the overall best performance for all datasets. How would the classification results change, if the authors used the best classifier for each dataset? This approach is better to assess the maximal information present in each dataset, without having to account for task-irrelevant differences in the datasets that could affect classification performance.

In 13/13 “before inactivation” datasets, the boosted decision tree (bDT) classifier yielded the highest correct classification rate. In 11/13 “during inactivation” datasets the bDT had the highest correct classification rate; in the remaining 2/13 “during inactivation datasets”, a naïve Bayes (NB) classifier had the highest correct classification rate. However, in both of these 2/13 cases, the NB correct classification rate “during inactivation” was significantly smaller than the “before inactivation” bDT correct classification rate, indicating that regardless of the classifier chosen, the information available for classification during same-side SC inactivation was always smaller than before same-side SC inactivation. For completeness, we now provide the relevant correct classification rates for the 2/13 sessions described above: session 1, before inactivation: bDT = 59% (897/1515; 95% CI = [57%, 62%]), during inactivation: bDT = 35% (913/2643; 95% CI = [33%, 36%]), NB: 42% (1114/2643; 95% CI = [40% 44%]); session 2, before inactivation bDT = 58% (1167/2015; 95% CI = [56%, 60%]), during inactivation: bDT = 35% (1085/3086; 95% CI = [33%, 37%]), NB = 43% (1338/3086; 95% CI = [42% 45%]). These are now reported in the Materials and methods section (subsection “Data analysis”).

– At a higher level, I don't get the rationale for testing how covert spatial attention changes the involvement of CDh in encoding non-spatial epoch information. A diagram/model showing the authors' assumptions would be helpful.

We tested how SC inactivation changes CDh neuronal encoding of non-spatial task-epoch information because we hypothesize that the SC provides CDh with signals related to both the location and timing of behaviorally relevant visual events. In our view, the information provided to CDh by SC would, in part, allow CDh neurons to be sensitive to particular spatial locations containing behaviorally relevant visual information (e.g. cue-side preferences). However, in addition to spatial information, the SC can also provide the time of occurrence of behaviorally relevant visual information – SC can report visual events to CDh. The hypothesis that SC provides timing information to CDh is well substantiated by the observation that the anatomical nodes by which SC may influence CDh have been demonstrated to carry signals reporting the occurrence of behaviorally relevant sensory events (Aosaki et al., 1994; Matsumoto et al., 2001). An example that we hope is clarifying: in our task, the “post-cue” epoch starts after spatial cue presentation; if same-side SC inactivation results in CDh neurons receiving erroneous information about the location and timing of relevant visual information, those neurons would be less sensitive to spatial cue presentation, and thus less sensitive to the transition from “pre-cue” to “post-cue” task-epochs. More generally, these concepts relate to our discussion of “belief states”, which allow an animal to keep track of ongoing environmental conditions and what those conditions imply both about appropriate ways to act in response to those conditions and about what environmental conditions are likely to occur next.

[Editors' note: further revisions were suggested prior to acceptance, as described below.]

Reviewer #1:This is a re-review of the study by Herman and colleagues on "Attention-related modulation of caudate neurons depends on superior colliculus activity". The authors have addressed many of the reviewers' concerns in satisfactory ways. My general main concern regarding the approach remains: First, it is difficult to interpret the effects of inactivating a structure on another structure without understanding the physiological interactions in the first place. Second, perturbing a part of a large-scale network will cause interruptions other than in the caudate nucleus, and these can influence or even account for what has been studied here. So, in the end, I am not clear what many of the results mean in a broader context. Still, given the scarcity of studies on the caudate and attention, I am supportive of publication, but I find the interpretations still overreaching including the title – perhaps a bit more caution might be helpful.

We agree that it is important to present a clear interpretation of the meaning of our results in a broader context, and to avoid unnecessary speculation. To that end, we made two modifications to the manuscript: (1) throughout the manuscript we point out that the observed effects of SC inactivation on CDh may be direct or indirect (Introduction, subsection “Effects of SC inactivation on CDh cue location preferences”, Discussion, subsection “A reversal of the classic subcortical hierarchy”); (2) we have added a paragraph to the Discussion section which acknowledges that our experiments do not speak to the specific circuit mechanism by which SC inactivation affects CDh (see the aforementioned subsection). This added paragraph also describes how the development of optogenetic tools would make it possible to perform experiments to disambiguate between different possible mechanisms by which SC output may influence CDh neurons.

Another point that remains and that needs to be addressed are the eye movement concerns raised by reviewer #2. Since the AUs have the data, it would be preferable to address the concern by showing compelling data rather than arguing that the influence of microsaccades is unlikely to play a role here. Microsaccades have been shown to play major roles in influencing attentional modulation, and I feel this issue can be addressed in better ways.

We agree, and have added analyses based on our eye movement recordings that rule out these concerns about microsaccades. We now show that changes in proportions of CDh neurons with cue-contra or cue-ipsi preferences during same-side SC inactivation, like those we present in Figure 4, are independent of microsaccades. We have included a new supplementary figure (Figure 4—figure supplement 3) that documents our microsaccade analysis approach and associated data analyses which we refer to in the Results section (subsection “Effects of SC inactivation on CDh cue location preferences”) and we describe our approach in the Materials and methods section (subsection “Data analysis”). Specifically, we recomputed the cue-side preference of each CDh neuron before and during SC inactivation in a limited subset of trials in which no microsaccades occurred (in the time window preceding and during each task-epoch). We determined these recomputed “no microsaccade” cue-side preferences exhibited the same pattern of changes during same-side SC inactivation as did cue-side preferences computed from all trials (that pattern being an increase in the proportion of cue-ipsi and decrease in cue-contra preferences).

Because we now provide this data, we have removed the paragraph of the Discussion section previously devoted to arguing for an unlikely influence of microsaccades on our results.

Reviewer #2:The authors adequately responded to all of my concerns. Although Figure 3B was lost in the main manuscript, I found it near the end of the manuscript as an individual figure. I recommend its publication in eLife.

We thank the reviewer for their support and for their keen eye in identifying the erroneously excluded Figure 3B.